# Retinoic acid signaling is directly activated in cardiomyocytes and protects mouse hearts from apoptosis after myocardial infarction

**Fabio Da Silva[1†], Fariba Jian Motamedi[1], Lahiru Chamara Weerasinghe Arachchige[1], Amelie Tison[1], Stephen T Bradford[1], Jonathan Lefebvre[1], Pascal Dolle[2], Norbert B Ghyselinck[2], Kay D Wagner[1], Andreas Schedl[1]\***

[1]Université Côte d'Azur, Inserm, CNRS, iBV, Nice, France; [2]IGBMC, Inserm U1258, UNISTRA CNRS, Illkirch, France

**Abstract** Retinoic acid (RA) is an essential signaling molecule for cardiac development and plays a protective role in the heart after myocardial infarction (MI). In both cases, the effect of RA signaling on cardiomyocytes, the principle cell type of the heart, has been reported to be indirect. Here we have developed an inducible murine transgenic RA-reporter line using CreER[T2] technology that permits lineage tracing of RA-responsive cells and faithfully recapitulates endogenous RA activity in multiple organs during embryonic development. Strikingly, we have observed a direct RA response in cardiomyocytes during mid-late gestation and after MI. Ablation of RA signaling through deletion of the *Aldh1a1/a2/a3* genes encoding RA-synthesizing enzymes leads to increased cardiomyocyte apoptosis in adults subjected to MI. RNA sequencing analysis reveals *Tgm2* and *Ace1*, two genes with well-established links to cardiac repair, as potential targets of RA signaling in primary cardiomyocytes, thereby providing novel links between the RA pathway and heart disease.

**\*For correspondence:**
Andreas.Schedl@unice.fr

**Present address:** [†]Deutsches Krebsforschungszentrum (DKFZ), Heidelberg, Im Neuenheimer, Heidelberg, Germany

**Competing interest:** The authors declare that no competing interests exist.

## Introduction

Retinoic acid (RA), the active derivative of vitamin A, plays essential roles in cell growth, differentiation, and organogenesis (*Duester, 2008*; *Cunningham and Duester, 2015*). RA is synthesized in two oxidative steps, with the second step being carried out by three retinaldehyde dehydrogenases (ALDH1A1, ALDH1A2, ALDH1A3). Once synthesized, RA can activate or repress the transcription of various genes by binding to nuclear retinoic acid receptors (RARA, RARB, RARG), which form heterodimers with retinoid X receptors (RXRA, RXRB, RXRG) (*Niederreither and Dollé, 2008*). Unlike most other embryonic signals that are peptidic in nature and act through membrane receptors, RA is a very small lipophilic molecule that cannot be easily detected by conventional means (*Rhinn and Dollé, 2012*). Hence, determining the activity of RA signaling in specific cell types or tissues is challenging, and has traditionally relied on RA reporter lines.

In mice, several transgenic RA reporter lines have been developed to better understand the RA signaling pathway. These include the *Tg(RARE-Hspa1b/lacZ)12Jrt* (hereafter referred to as RARE-LacZ), *Tg(RARE/Hspa1b-cre)1Dll* (referred to as RARE-Cre) and RARE-Luciferase lines, among others (*Rossant et al., 1991*; *Dollé et al., 2010*; *Bilbija et al., 2012*). All three lines utilize multiple copies of the *Rarb* retinoic acid response element (RARE) to drive β-galactosidase (β-gal), Cre recombinase, or Luciferase expression, and they have proven useful in understanding the wide range of RA responses during embryonic development and disease. Yet, due to inherent technical problems, these lines are

limited in their ability to illustrate the full scope of RA activity in the mouse. For instance, the RARE-LacZ line is highly efficient at detecting acute RA activity at various stages of embryonic development, but is unable to trace the long-term fate of RA-responsive cells (*Rossant et al., 1991*). Moreover, the β-gal protein is very stable and persists in a tissue long after its expression ceases, so measurement of its activity may not reflect actual RA signaling. The RARE-Cre is also very effective at detecting RA-responses in tissues, but since it promotes constitutive Cre expression and permanent labeling of cells, the timing of the response cannot be determined (*Dollé et al., 2010*). The RARE-Luciferase line can detect RA activity in vivo, but due to its use of Luciferase as a reporter, the cell types responding to RA in specific tissues cannot be identified by conventional immunohistochemistry (*Bilbija et al., 2012*). Hence, despite the wide variety of lines available, there remains a need for new tools capable of detecting RA activity in a controlled, reliable, and efficient manner in order to better understand the roles of RA signaling in complex organs.

One such organ with which RA signaling is intimately linked is the heart. In order to meet high metabolic demands and ensure enough blood is efficiently pumped throughout the body, the mammalian heart must develop a compact layer of cardiomyocytes, whose coordinated contractions drive the pumping action of the heart (*Zhang et al., 2013*). Myocardial compaction occurs between embryonic day (E)10–14 in the mouse, and involves the proliferation and differentiation of cardiomyoblasts, the precursors of cardiomyocytes (*Zaffran et al., 2014*; *Xavier-Neto et al., 2015*; *Nakajima, 2019*). RA signaling is intricately linked with myocardial compaction and both *Rxra*-null and *Aldh1a2*-null mutant mice display severe hypoplasia of the compact layer (*Jenkins et al., 2005*; *Merki et al., 2005*; *Lin et al., 2010*). Despite having a pronounced effect on the myocardium, it has been suggested that the effects of RA signaling on developing cardiomyocytes is indirect. For instance, studies have shown that RA, synthesized by ALDH1A2 in the epicardium, works in an autocrine manner with RXRA functioning as the principal receptor (*Zaffran et al., 2014*; *Nakajima, 2019*). The ALDH1A2/RXRA axis stimulates the production of mitogens such as FGFs, which are then secreted to the myocardium to promote cell proliferation (*Tran and Sucov, 1998*; *Stuckmann et al., 2003*; *Lavine et al., 2005*). It has also been suggested that ALDH1A2/RXRA signaling in the liver and placenta promote the production of erythropoietin (EPO) and distribution of glucose, respectively, both of which sequentially activate epicardial *Igf2* expression (*Brade et al., 2011*; *Shen et al., 2015*). IGF2 then stimulates myocardial proliferation and compaction. Hence, whether the effects of retinoids are within the epicardium or extra-cardiac tissues (or both), it appears that RA signaling does not act directly in cardiomyocytes.

Regarding mammalian heart disease, many studies conducted in adult rats and mice suggest that RA signaling stimulates cardiac repair after myocardial infarction. Vitamin A-deficient rats subjected to ligation of the left anterior descending artery (LAD), a model of MI, exhibit increased cardiomyocyte hypertrophy and interstitial collagen deposition, while supplementation of RA to mice following ischaemia/reperfusion surgery results in decreased apoptosis and smaller infarct zones (*Minicucci et al., 2010*; *Zhu et al., 2015*). In both cases, the cell types responding to RA treatment were not identified. Direct evidence that RA signaling is reactivated in the heart post MI comes from a study performed with the RARE-Luciferase reporter line. In this study, the authors detected RA signaling in damaged hearts 24 hr to 1 week post MI (*Bilbija et al., 2012*). Cardiac fibroblasts, and not cardiomyocytes, were determined to be the principal cell-types responding to RA, once again suggesting an indirect effect of RA on the myocardium.

Despite the majority of studies suggesting that cardiomyocytes are not the major RA-responsive cell types in the heart, minor, albeit important roles for RA signaling in the mammalian myocardium have been identified. In a study by *Guleria et al., 2011*, it was demonstrated that stimulation of the RA pathway in cultured neonatal cardiomyocytes subjected to high-glucose conditions led to decreased apoptosis. The protective effect was attributed to RA-induced modulation of the Renin-angiotensin system (RAS). More recently, cardiomyocyte-specific deletion of the RARA receptor in adult mice led to increased cardiomyocyte hypertrophy, excessive reactive oxygen species (ROS) accumulation and calcium mishandling defects (*Zhu et al., 2016*). Furthermore, in non-mammalian vertebrates capable of naturally restoring damaged cardiac tissue without scar formation, it has been shown that RA signaling is indispensable for the regenerative response (*Fernandez et al., 2018*). For example, in adult zebrafish subjected to ventricular resection injuries, *Aldh1a2* mRNA was found to be upregulated in the endocardium and epicardium, and inhibition of RA signaling through transgenic expression of dominant-negative RARA led to drastically reduced cardiomyocyte proliferation

(*Kikuchi et al., 2011*). Altogether, these studies suggest RA signaling is active in cardiomyocytes, and that the RA pathway may be involved in protecting and/or repairing damaged heart muscle.

Here, we have developed a novel RA reporter line using CreER$^{T2}$ technology (RARECreER$^{T2}$) that closely mimics well-known RA reporter lines and allows for timed, cell-type-specific tracing of RA-responsive cells. Using this line, we have detected a myocardial-specific response to RA during mid-late stages of development (E11-18), and have observed RA activity in cardiomyocytes of adult mice subjected to MI. In order to better understand the role of RA in cardiac repair, we have crossed *loxP*-flanked (floxed) alleles of the *Aldh1a* enzymes (*Matt et al., 2005*; *Vermot et al., 2006*; *Dupé et al., 2003*) with the ubiquitously expressed and inducible *Tg(CAG-cre/Esr1\*)5Amc* line, hereafter referred to as CAGGCreER (*Hayashi and McMahon, 2002*), to generate time-controlled loss of functions. By temporally deleting the *Aldh1a* alleles in adult mice then subjecting them to MI, we have observed a drastic increase in cardiomyocyte-specific apoptosis. RNA sequencing of primary cardiomyocytes treated with RA followed by genome-wide analysis reveals RA can stimulate a notable transcriptional response in cardiomyocytes. Importantly, some of the identified RA-responsive genes, such as transglutaminase 2 (*Tgm2*) and angiotensin converting enzyme (*Ace1*), have well characterized roles in cardiac repair, thereby providing new molecular links between RA signaling and heart disease.

## Results

### A novel RARECreER$^{T2}$ line recapitulates endogenous RA signaling in developing mouse embryos

The RA signaling pathway is extremely dynamic, acting on various tissues at different stages of embryonic development (*Duester, 2008*; *Niederreither and Dollé, 2008*). In order to better understand the timing and identity of the cell types responding to RA, we developed a novel tamoxifen-inducible RA reporter line using CreER$^{T2}$ technology (*Figure 1A*, schematic). Previous work has shown that RA signaling, activated by expression of *Aldh1a2* in caudal regions of the embryo, is first detected around E7.5 (*Rossant et al., 1991*, *Ang et al., 1996*). To see if we could detect this early RA activity, we crossed our line with the *Gt(ROSA)26Sor^{tm1Sor}* reporter line (referred to as R26L) (*Soriano, 1999*) and administered tamoxifen (TAM) one day earlier, at E6.5, to account for tamoxifen processing and eventual reporter activation (*Figure 1A–B*). Strikingly, analysis of embryos at E9.5 via whole-mount X-gal staining demonstrated high reporter activity up to posterior hindbrain/pharyngeal regions of the embryo (*Figure 1B*, left panel). Administration of tamoxifen at E7.5 followed by analysis at E10.5 revealed a similar pattern of staining, as well as efficient labeling of the developing limbs (*Figure 1B*, right panel). The X-gal staining at both time-points closely matched the staining pattern observed with other RARE reporter lines such as the RARE-LacZ and RARE-Cre, and is consistent with the conserved roles of RA signaling role in embryo posteriorization, and limb bud formation (*Rossant et al., 1991*; *Dollé et al., 2010*; *Durston et al., 1989*; *Sive et al., 1990*; *Stratford et al., 1999*; *Niederreither et al., 2002b*). However, we detected very few X-gal$^+$ cells in the forebrains of our RARECreER$^{T2}$ embryos, a known site of RA activity (*Rossant et al., 1991*; *Dollé et al., 2010*; *Ribes et al., 2006*). To address this issue, we crossed our line with the *Gt(ROSA)26Sor^{tm4(ACTB-tdTomato,-EGFP)Luo}* reporter line (referred to as mTmG) (*Muzumdar et al., 2007*), which expresses GFP protein upon Cre recombination and is more sensitive than the R26L line. Whole mount GFP immunofluorescence (IF) of embryos induced at E7.5 and sacrificed at E10.5 revealed efficient labeling of the forebrain (*Figure 1C* and *Video 1*). To examine the full spectrum of the RA response detected with the RARECreER$^{T2}$ line, we repeated the lineage-tracing experiments with two pulses of tamoxifen at E7.5 and E8.5 to improve the overall recombination efficiency, and then analyzed embryos at E11.5, when most organs are further developed. Analysis of whole embryo sections by GFP IF revealed efficient and consistent labeling of various well-known RA-responsive organs such as the lungs (*Malpel et al., 2000*), liver (*Brade et al., 2011*), spinal cord (*Sockanathan and Jessell, 1998*), heart (*Niederreither et al., 2001*; *Niederreither et al., 2002a*), and differentiating somites (*Rossant et al., 1991*; *Niederreither et al., 1999*; *Diez del Corral et al., 2003*; *Duester, 2007*) in all embryos analyzed (*Figure 1D*). Somite-specific labeling, as revealed by co-IF for MyoD and GFP, was also detected with a single pulse at E8.5 followed by analysis at E11.5 (*Figure 1E*).

Another organ with a very well-characterized RA response is the eye. *Aldh1a2* is detected in the optic vesicle as early as E8.5, where it is required for the transition from optic vesicle to optic cup

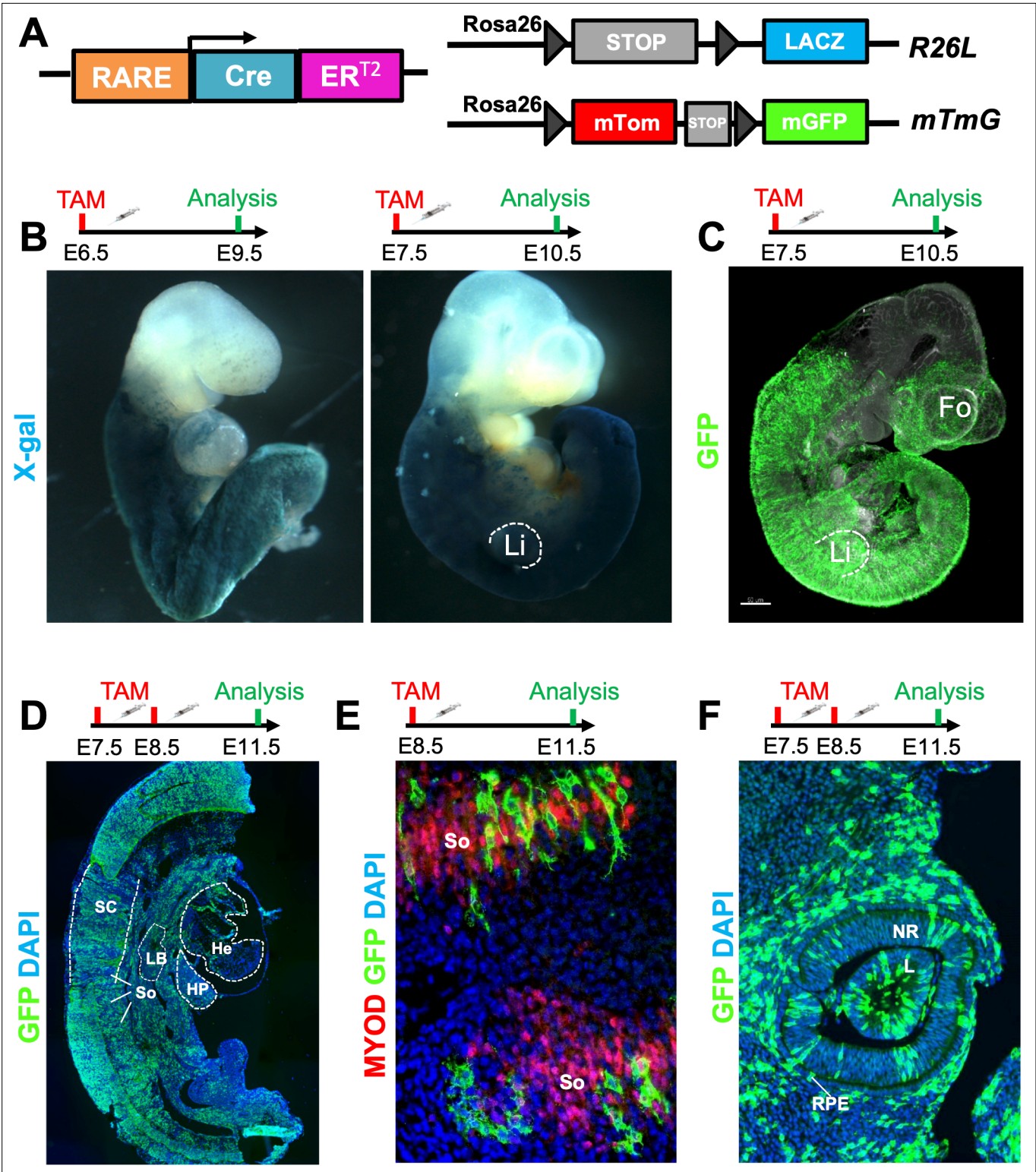

**Figure 1.** A novel RARECreER[T2] line recapitulates endogenous RA signaling in mouse embryos. (**A**) Scheme illustrating the strategy used to test the novel RARECreER[T2] line during various stages of embryonic development. The RARECreER[T2] line was crossed with the *Rosa26LacZ* (R26L) or the membrane targeted tandem dimer Tomato membrane targeted green fluorescent protein (mTmG) reporter lines and recombination was induced via tamoxifen (TAM) administration at multiple time-points. Gray arrowheads represent *lox*P sites. (**B**) Whole-mount X-gal staining of RARECreER[T2] R26L embryos induced at embryonic day 6.5 (E6.5) or E7.5 followed by respective analyses at E9.5 or E10.5 demonstrates efficient labeling with the *lacZ* reporter. Notice the labeling of the forelimbs (Li) with the E7.5 pulse. (**C**) Whole-mount GFP IF of RARECreER[T2] embryos crossed with the mTmG

*Figure 1 continued on next page*

*Figure 1 continued*

reporter, pulsed with tamoxifen at E7.5 and sacrificed at E10.5 reveals specific forebrain (Fo) labeling. (**D**) GFP IF on sagittal sections of E11.5 embryos pulsed with tamoxifen at E7.5 and E8.5. Labeling is detected in various organs and tissues with well described RA activity such as the heart (He), spinal cord (SC), hepatic primordium (HP), lung bud (LB), and developing somites (So). Heads were removed for analysis. (**E**) Co-IF with GFP and Myoblast determination protein (MYOD) antibodies reveals RA-responsive cells in developing somites when pulsed at E8.5 and analyzed at E11.5. (**F**) Embryos pulsed with tamoxifen at E7.5 and E8.5 and analyzed at E11.5 display efficient labeling of the neural retina (NR), lens (L) and retinal pigment epithelium (RPE) of the eye as determined by GFP IF. Data information: For all experiments, n = 3 embryos were analyzed and representative embryos shown. See also *Figure 1—figure supplement 1*.

The online version of this article includes the following figure supplement(s) for figure 1:

**Figure supplement 1.** The RARECreER$^{T2}$ line points to a reduced response at later time-points in embryonic development.

(*Niederreither et al., 1997*; *Wagner et al., 2000*; *Mic et al., 2004*). As development proceeds, *Aldh1a1* expression in the dorsal retina and *Aldh1a3* expression in the surface ectoderm, lens placode, retinal pigment epithelium and, later, the ventral retina, continue to activate RA signaling in the eye (*Matt et al., 2005*; *Dupé et al., 2003*; *McCaffery et al., 1991*; *Mic et al., 2000*). Analysis of RARECreER$^{T2}$ embryos pulsed with tamoxifen at E7.5 and E8.5 and analyzed at E11.5 revealed efficient GFP labeling of the lens placode, neural retina, and prospective retinal pigment epithelium cells, consistent with previously published data (*Figure 1F*; *Mic et al., 2004*). Embryos pulsed at E8.5 and analyzed at E11.5 also displayed labeling in the eye, although at a lower frequency (*Figure 1—figure supplement 1A*). These results are consistent with the fact that RA activity in the eye is continuous during initial stages of its development, since multiple pulses of tamoxifen lead to more efficient labeling when compared to only one shot of tamoxifen. Analysis of embryos pulsed at E8.5 and analyzed either at E10.5 (R26L) or E11.5 (mTmG) revealed labeling in the eyes, forebrain, heart, and somites, but the overall labeling efficiency was decreased when compared to earlier time-points (*Figure 1—figure supplement 1B-C*). Tamoxifen administration at E10.5 followed by light sheet microscopy analysis of whole embryos at E12.5 revealed a similar GFP labeling pattern as the E8.5 pulses (*Figure 1—figure supplement 1D*). In summary, the RARECreER$^{T2}$ line efficiently labels organs and tissues with well described RA activity in a highly dynamic fashion, and its labeling efficiency peaks during early stages of embryogenesis.

## Cardiomyocytes are highly responsive to RA signaling during embryonic development

RA signaling is essential for cardiac formation and displays unique spatiotemporal patterns of activity in the heart (*Ryckebusch et al., 2008*; *Xavier-Neto et al., 2015*; *Nakajima, 2019*). During early stages of cardiac development, RA signaling is active in the venous pole of the heart. At later time-points, RA activity relocates to the heart's outer layer, otherwise known as the epicardium (*Moss et al., 1998*; *Xavier-Neto et al., 2000*). A closer look at the hearts of E9.5 RARECreER$^{T2}$ embryos stained with X-gal after TAM treatment at E6.5 revealed specific labeling of the venous pole derivatives (atria and outflow tract) (*Figure 2A* and *Figure 2—figure supplement 1A*), and minimal labeling of the developing ventricles, faithfully recapitulating the pattern observed with other RA reporters (*Rossant et al., 1991*; *Dollé et al., 2010*). Meanwhile, embryos pulsed with tamoxifen at E10.5 and analyzed at E13.5 exhibited very strong labeling of the heart ventricles with almost no labeling of the atria and outflow tract (*Figure 2B*), once again consistent with previously published data (*Moss et al., 1998*). Interestingly, analysis of E13.5 heart sections counterstained with eosin revealed the majority of the X-gal labeling to be within the myocardial wall of the heart and not the epicardium (*Figure 2—figure supplement 1B*).

To more precisely determine the cell-types labeled by the RARECreER$^{T2}$ line during mid stages of gestation, we conducted lineage tracing

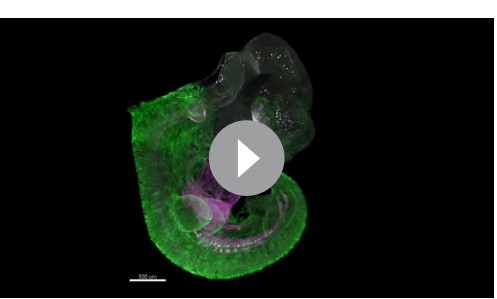

**Video 1.** The RARECreER$^{T2}$ line displays a high labeling efficiency in early mouse embryos. Wholemount IF for GFP (green) and Gata binding protein 4 (Gata4)(purple) on E10.5 RARECreER$^{T2}$; mTmG embryos injected with tamoxifen at E7.5.
https://elifesciences.org/articles/68280/figures#video1

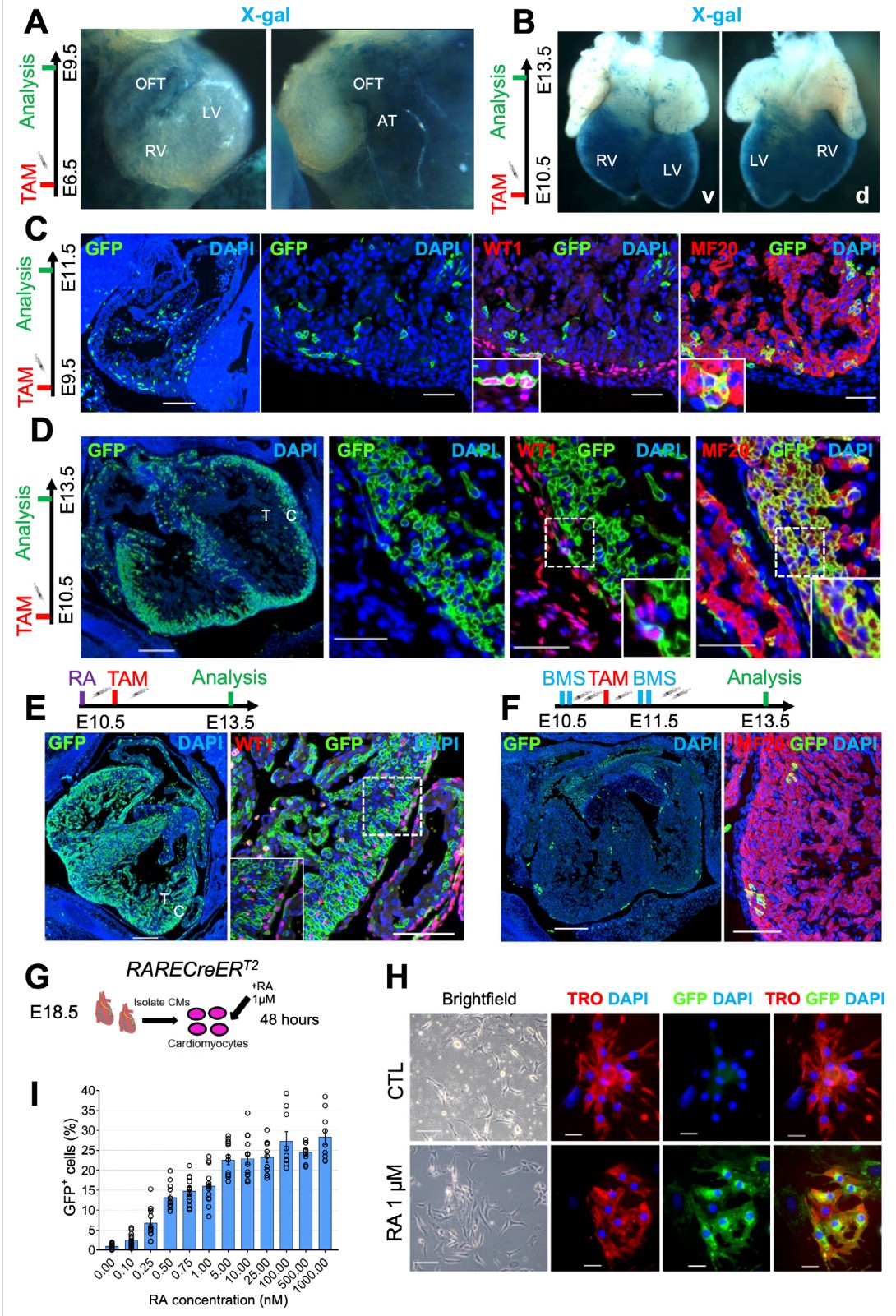

**Figure 2.** Cardiomyocytes are highly responsive to RA signaling during embryonic development. (**A**) Whole-mount X-gal staining of RARECreER[T2] embryos pulsed with tamoxifen at E6.5 and sacrificed at E9.5 reveals strong labeling of venous pole derivatives of the heart (outflow tract (OFT) and atria (AT)). Minimal labeling is detected in the right ventricle (RV) and left ventricle (LV) of hearts (n = 2 embryos analyzed). (**B**) Whole-mount X-gal staining of embryos pulsed with tamoxifen at E10.5 and sacrificed at E13.5 reveals strong ventral (v) and dorsal (d) labeling of the heart ventricles with

*Figure 2 continued on next page*

*Figure 2 continued*

minimal labeling of the atria and outflow tract (n = 2 embryos analyzed). (**C**) Administration of tamoxifen (TAM) at E9.5 to RARECreER^T2; mTmG embryos followed by analysis at E11.5 reveals specific labeling of the epicardium (GFP⁺WT1⁺ cells) and myocardium (GFP⁺MF20⁺ cells) in developing hearts (n = 3 embryos analyzed). (**D**) Tamoxifen administration at E10.5 followed by analysis at E13.5 reveals strong labeling of the compact myocardium (C) and minor labeling of the trabecular layer (T) in developing hearts. Minimal labeling of the epicardium (WT1⁺ cells, inset) is detected at this time-point. Insets shown are from right ventricle of representative heart (n = 3 embryos analyzed). (**E**) Exogenous supplementation of all-trans Retinoic acid (RA) (10 mg/kg) to pregnant dams 4 hr prior to tamoxifen induction leads to increased labeling of the trabecular myocardium and epicardium (WT1⁺ cells, inset) in developing hearts at E13.5 when compared to non-RA-treated embryos in (**D**) (n = 3 embryos analyzed). (**F**) Supplementation of the RAR reverse agonist BMS493 (5 mg/kg) to pregnant dams 4 hr before and 4 hr after tamoxifen induction (extra two doses given in 8 hour interval 1 day after TAM induction) drastically reduces the number of GFP⁺ cells in RARECreER^T2 hearts (n = 3 embryos analyzed). (**G**) Schematic illustrating strategy for isolating primary cardiomyocytes (CMs) from hearts of E18.5 RARECreER^T2; mTmG embryos followed by 48 hr treatment with 1 µM RA. (**H**) RARECreER^T2; mTmG primary cardiomyocytes respond directly to RA treatment as demonstrated by co-IF for GFP and Troponin T. No GFP staining is detected in DMSO-treated control (CTL) cells. (**I**) Dose-response relationship of primary cardiomyocytes to RA. Cells were isolated form neonatal RARECreER^T2; mTmG hearts using the neonatal cardiomyocyte isolation kit (Miltenyi) and treated for 48 hr with varying doses of RA. Each data point represents the percentage of GFP⁺ cells from a single well. Columns are means ± SEM (at least nine technical replicates per treatment). See also *Figure 2—source data 1*. Data information: WT1 = Wilms' tumour protein, TRO = Troponin T, MF20 = Myosin heavy chain. Scale bars mosaics: 100 µM, Close ups: 40 µM. See also *Figure 2—figure supplement 1*.

The online version of this article includes the following figure supplement(s) for figure 2:

**Source data 1.** Numerical source data for *Figure 2I*.

**Figure supplement 1.** The RARECreER^T2 line is active in the venous pole of the heart during early stages of development and in the myocardium during later stages, and myocardial labeling during mid-gestation is detected with the RARECreER^T2 line B.

experiments with the mTmG reporter allele, and then performed co-IF with GFP and epicardial (WT1 (Wilms' tumour protein)) or myocardial (MF20 (myosin heavy chain))-specific antibodies. Administration of tamoxifen at E9.5 followed by analysis at E11.5 revealed GFP localization in both WT1⁺ and MF20⁺ cells (*Figure 2C*). Meanwhile, tamoxifen administration at E10.5 followed by analysis at E13.5 revealed high labeling of MF20⁺ cardiomyocytes, but only minor labeling of WT1⁺ epicardial cells (*Figure 2D*). The cardiomyocyte-specific labeling was highest in the compact layer, with minimal GFP⁺ cells observed in trabecular cardiomyocytes (*Figure 2D*, left panel). Taken together, these data demonstrate that as heart development proceeds, cardiomyocyte precursors of the compact layer increasingly become the major RA-responsive cell-type.

To validate that the activation of GFP in cardiomyocytes from our RARECreER^T2 embryos reflects true RA signaling, we performed various in vivo and in vitro experiments. Administration of RA to pregnant dams at E10.5 4 hr prior to TAM induction increased the RA-response in RARECreER^T2 hearts analyzed at E13.5, even labeling a substantial portion of trabecular myocytes (*Figure 2E*). Conversely, treatment of embryos with the RAR inverse agonist BMS493 (*Germain et al., 2009*) prior to and following tamoxifen administration led to a very strong decrease in GFP⁺ cells (*Figure 2F*). Moreover, isolation of primary cardiomyocytes from E18.5 RARECreER^T2 hearts followed by RA treatment led to specific labeling of Troponin T⁺ cardiomyocytes (*Figure 2G*, schematic and *Figure 2H*). The efficiency of the RA treatment in cultured cardiomyocytes was confirmed by qPCR analysis, which showed upregulation of the RA target genes *Rarb*, *Rbp1* (retinol binding protein), and *Cyp26a1* (cytochrome P450 family 26 subfamily a polypeptide 1) (*Figure 2—figure supplement 1C*).

To better characterize the dynamics of RA signaling in cardiomyocytes, we performed time-course and dose titration experiments in cultured cells. First, we treated primary cardiomyocytes isolated from RARECreER^T2; mTmG hearts with 10 nM RA for different periods of time (8–48 hr) and found that GFP⁺ cells could be detected as early as 28 hr, with peak levels at 48 hr. Next, we treated cardiomyocytes with increasing doses of RA (0.1–1000 ng/ml) for 48 hr and quantified the number of GFP⁺ cells. Treated cardiomyocytes showed a dose-dependent response to RA (*Figure 2I* and *Figure 2—figure supplement 1D*). Strikingly, GFP⁺ cells could also be detected with very low doses of RA (0.1 nM – 1 nM) (*Figure 2I* and *Figure 2—figure supplement 1D*). Taken together, these data demonstrate that cardiomyocytes are highly responsive to exogenous RA treatment at physiologically relevant levels.

Finally, to ensure the in vivo cardiomyocyte response was not an artefact of our transgenic line, we performed lineage-tracing experiments with a second RARECreER^T2 line (Line B). Administration of tamoxifen at E10.5 followed by analysis at E14.5 revealed a nearly identical pattern of GFP staining in the compact myocardium when compared to our original line (*Figure 2—figure supplement 1E*). Altogether, these data demonstrate that the cardiomyocyte-specific labeling observed with the

RARECreER[T2] line is a reliable response that can be influenced by exogenous activation or inactivation of the RA pathway.

## Cardiomyocyte-specific RA signaling is active during late stages of heart development

Although the role of RA signaling during heart looping and myocardial compaction is well described, it is unclear whether retinoids continue to play an important role after E13.5 (*Zaffran et al., 2014*). We next decided to look at later time-points of heart development by administering tamoxifen at E14.5 and sacrificing embryos at E18.5. Strikingly, we observed a high number of GFP[+] cells, mainly cardiomyocytes (*Figure 3A*). We also noticed that many of the labeled cells were located deep within the compact layer (*Figure 3A*, white arrows), a striking observation given that at these time-points ALDH1A2 protein is believed to be restricted to the epicardium (*Moss et al., 1998*). To see if this deep labeling was simply due to clonal expansion of cardiomyocytes located near the epicardium, we traced cells for a shorter period of time by giving a pulse at E15.5 and analyzing at E17.5. Once again, we observed cardiomyocyte-specific labeling, both near the epicardium as well as deeper within the myocardial wall (*Figure 3B*). Many of the GFP[+] cardiomyocytes located away from the epicardium were separated from other cells (*Figure 3B*, white arrows), suggesting their response arises from a local source of RA, independently of the epicardium.

During cardiac formation, ALDH1A2 is the main enzyme responsible for RA synthesis and its loss of function leads to cardiac defects at both early and mid-stages of development (*Niederreither et al., 2001*; *Lin et al., 2010*; *Braitsch et al., 2012*). Whether or not ALDH1A2 has a role at later time-points is unknown since full *Aldh1a2*-null embryos as well as *Aldh1a2*-null embryos rescued by a short-term RA supplementation die at E9.5 and E13.5, respectively (*Niederreither et al., 2001*; *Lin et al., 2010*). To see if the RA response detected after E13.5 was linked to *Aldh1a2* expression, we performed IF with an anti-ALDH1A2 antibody at various stages of gestation. At E12.5 ALDH1A2 was, as previously shown, restricted to the epicardium. After E14.5, however, ALDH1A2 could be detected within the ventricular and interventricular walls. This expression pattern increased at later time-points, peaking at E18.5 (*Figure 3C*). To determine which cell-types were positive for ALDH1A2 we performed co-IF with various antibodies. No co-IF of ALDH1A2 with cardiomyocyte (MF20) (data not shown), smooth muscle (Transgelin) (data not shown) or endothelial (PECAM1) (*Figure 3D*) markers was detected. By contrast, ALDH1A2[+] cells were positive for the intermediate filament marker Vimentin, suggesting them to be cardiac fibroblasts or connective tissue (*Figure 3D*). We next wanted to see if the ALDH1A2[+] cells arose from the epicardium, which would be consistent with them being fibroblasts or other connective tissue cells. To this end, we performed lineage tracing experiments with the *Wt1^tm2(cre/ERT2)Wtp* line (referred to as WT1CreER[T2]) (*Zhou et al., 2008*), which specifically labels the epicardium, crossed with the mTmG reporter. Tamoxifen induction at E10.5 followed by analysis at E16.5 revealed a significant portion of ALDH1A2[+] cells within the ventricular wall were labeled with GFP, demonstrating they indeed originate from the epicardium (*Figure 3E*, white arrows). Taken together, these data suggest that epicardial cells and epicardial-derived cardiac fibroblasts that have migrated into deeper layers of the forming myocardium produce ALDH1A2, and that the local RA signal is received by neighboring cardiomyocytes (*Figure 3F*, schematic).

## The RARECreER[T2] line labels several cell-types, including cardiomyocytes, in adult hearts subjected to myocardial infarction

RA signaling appears to play a protective role in the adult heart after acute damage (*Minicucci et al., 2010*; *Zhu et al., 2015*; *Paiva et al., 2005*). To further analyze RA signaling in cardiac repair, we subjected RARECreER[T2] mice to surgical ligation of the left coronary artery (a widely used experimental myocardial infarction model). To label RA-responsive cells, one pulse of tamoxifen was given immediately after surgery, and a second pulse 48 hr later (*Figure 4A*, schematic). Strikingly, analysis of hearts 6 days after MI revealed drastic GFP enrichment in infarct hearts when compared to sham controls (*Figure 4B*). ALDH1A2 protein and mRNA levels were also enriched in infarcted hearts as shown by IF and qPCR analysis, respectively (*Figure 4B–C*). Interestingly, the staining pattern of GFP closely overlapped with ALDH1A2, suggesting it to be the major enzyme driving the RA response in infarct hearts, which is consistent with previous studies (*Kikuchi et al., 2011*). *Aldh1a1* and *Aldh1a3* mRNA levels were also upregulated as determined by qPCR analysis (*Figure 4C*); however, we could

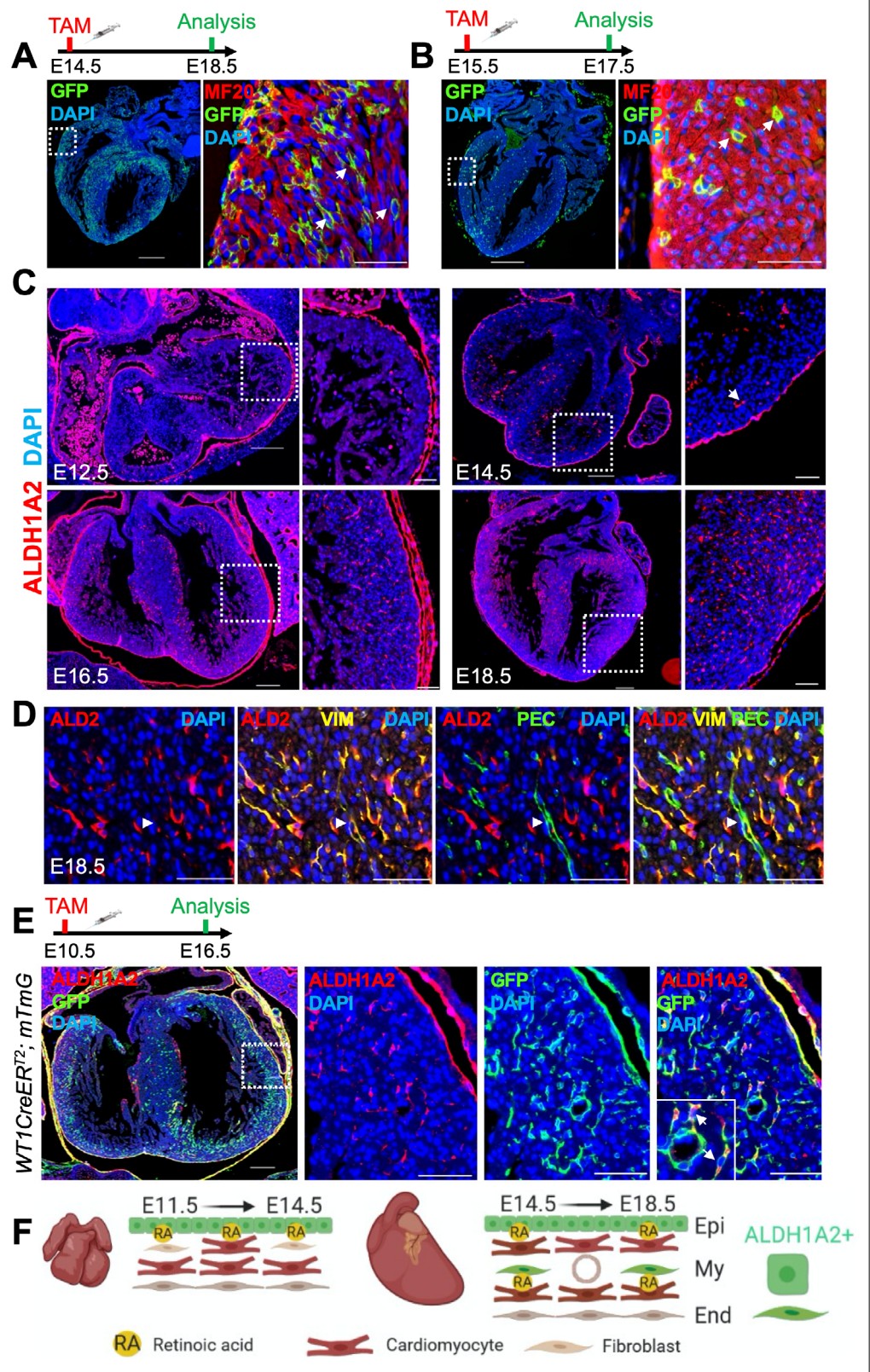

**Figure 3.** Cardiomyocyte-specific RA signaling is active during late stages of heart development. (**A**) Administration of tamoxifen (TAM) to RARECreER[T2]; mTmG embryos at E14.5 followed by analysis at E18.5 reveals cardiomyocyte-specific (MF20[+]) labeling in developing hearts. Cardiomyocytes located deep within the ventricular wall are also labeled (white arrows). (**B**) Tamoxifen administration at E15.5 labels cardiomyocytes deep within the

*Figure 3 continued on next page*

*Figure 3 continued*

ventricular wall (white arrows) when analyzed only 2 days later at E17.5. (**C**) IF analysis reveals ALDH1A2 protein is restricted to the epicardium of the developing heart at E12.5. At E14.5 ALDH1A2 protein is detected within the ventricular wall (white arrow). High ALDH1A2 protein levels are detected in the ventricular wall at E16.5 and E18.5 (at least three embryos analyzed per time-point). (**D**) Co-IF with Vimentin (VIM) and PECAM (PEC) antibodies reveals ALDH1A2 (ALD2) is produced by cardiac fibroblasts/connective tissue (VIM+) and not by endothelial cells (VIM+PEC+, white arrowheads) in the developing heart. Images taken from representative region of interventricular septum. (**E**) Administration of tamoxifen to embryos carrying the WT1CreER$^{T2}$ (epicardial-specific CreER$^{T2}$ line) and mTmG alleles followed by analysis at E16.5 reveals many of the ALDH1A2+ cells within the ventricular wall of the developing heart are derived from the epicardium (GFP+). (**F**) Scheme illustrating pattern of ALDH1A2 production and RA activity in the myocardium during mid-late stages of cardiac development. Image created with Biorender software. Epi = epicardium, My = myocardium, End = endocardium. Data information: Scale bars mosaics: 100 µM, Close ups: 40 µM. For all experiments at least three hearts were analyzed and representative hearts shown.

not detect ALDH1A1/A3 protein by IF using commercial antibodies (data not shown). A closer look at the staining patterns of ALDH1A2 and GFP revealed an increase in the number of positive cells particularly within the injury and border zones of damage (*Figure 4B, F*). Most ALDH1A2+ cells were GFP-negative (*Figure 4D*). Co-IF for GFP with various markers demonstrated that many different cell-types were responsive to RA. These included PECAM1+ coronary vessels, αSMA+ activated fibroblasts and/or smooth muscle cells and Troponin T+ cardiomyocytes (*Figure 4E*). Interestingly, a substantial portion of cardiomyocytes in the injury border zone, a region that is highly remodeled after MI, were also GFP+ (*Figure 4G*, white arrows).

## Depletion of RA signaling leads to larger infarct zones and increased apoptosis

Since our RARECreER$^{T2}$ line showed a strong response in regions of the heart that are highly remodeled during cardiac repair, we next wanted to study the effects of MI on mice with reduced levels of RA signaling. Deletion of *Aldh1a1/a2/a3* with the CAGGCreER line at E13.5, when *Aldh1a2* is actively expressed, revealed drastically reduced staining for ALDH1A2 (*Figure 5—figure supplement 1A,B*) and qPCR analysis confirmed a very high ( > 90%) deletion efficiency (*Figure 5—figure supplement 1C*) in E18.5 hearts of triple *Aldh1a*-knockout mice (hereafter referred to as RAKO). This developmental analysis confirmed the efficacy of our approach and we decided to use the same Cre line for the *Aldh1a* deletion in adults. A previous report has shown that activation of the CAGGCreER transgene alone by tamoxifen injection followed by MI surgery does not lead to adverse remodeling effects when compared to control hearts without Cre (*Konstandin et al., 2013*). As control mice for MI experiments in adults, we therefore used floxed *Aldh1a1/a2/a3* mice without the CAGGCreER transgene. To delete the enzymes, we injected mice five times with tamoxifen one week prior to surgery (*Figure 5A*, schematic). IF analysis demonstrated decreased ALDH1A2 immunoreactivity in RAKO hearts 6 days post MI (*Figure 5B*), and qPCR analysis revealed reasonable deletion efficiency, with around 70 % decrease of *Aldh1a1* and *Aldh1a2* mRNA levels (*Figure 5C*). *Aldh1a3* mRNA levels were not significantly downregulated (*Figure 5C*). Incomplete deletion of *Aldh1a* genes was likely due to epigenetic silencing of their loci, since they are not actively expressed in adult hearts prior to MI (*Kikuchi et al., 2011*; this study). Consistently, PCR analysis of genomic DNA from mutant MI hearts revealed incomplete excision of *Aldh1a2* and *Aldh1a3* floxed alleles (*Figure 5—figure supplement 1D-E*). The low recombination was not due to poor activity of the CAGGCreER line, which, upon crossing with the *Rosa26* based mTmG reporter, showed very efficient recombination in the adult heart (*Figure 5—figure supplement 1F-G*).

To analyze the effects of depleting the RA pathway prior to MI, we measured the size of the infarct zones in RAKO and control mice. Strikingly, RAKO mice exhibited significantly increased infarct zones as revealed by collagen staining with Sirius red (*Figure 5D–E* and *Figure 5—figure supplement 2A*). Furthermore, RAKO mice showed increased rates of apoptosis as determined by active caspase three and TUNEL (Terminal deoxynucleotidyl transferase dUTP nick end labeling) staining (*Figure 5F–G*). No apoptosis was observed in RAKO mice subjected to sham surgery (*Figure 5—figure supplement 2B*). Interestingly, many of the apoptotic cells in RAKO mice were cardiomyocytes and they tended to concentrate in clusters or so called 'patches' that were positive for both active caspase3 and TUNEL

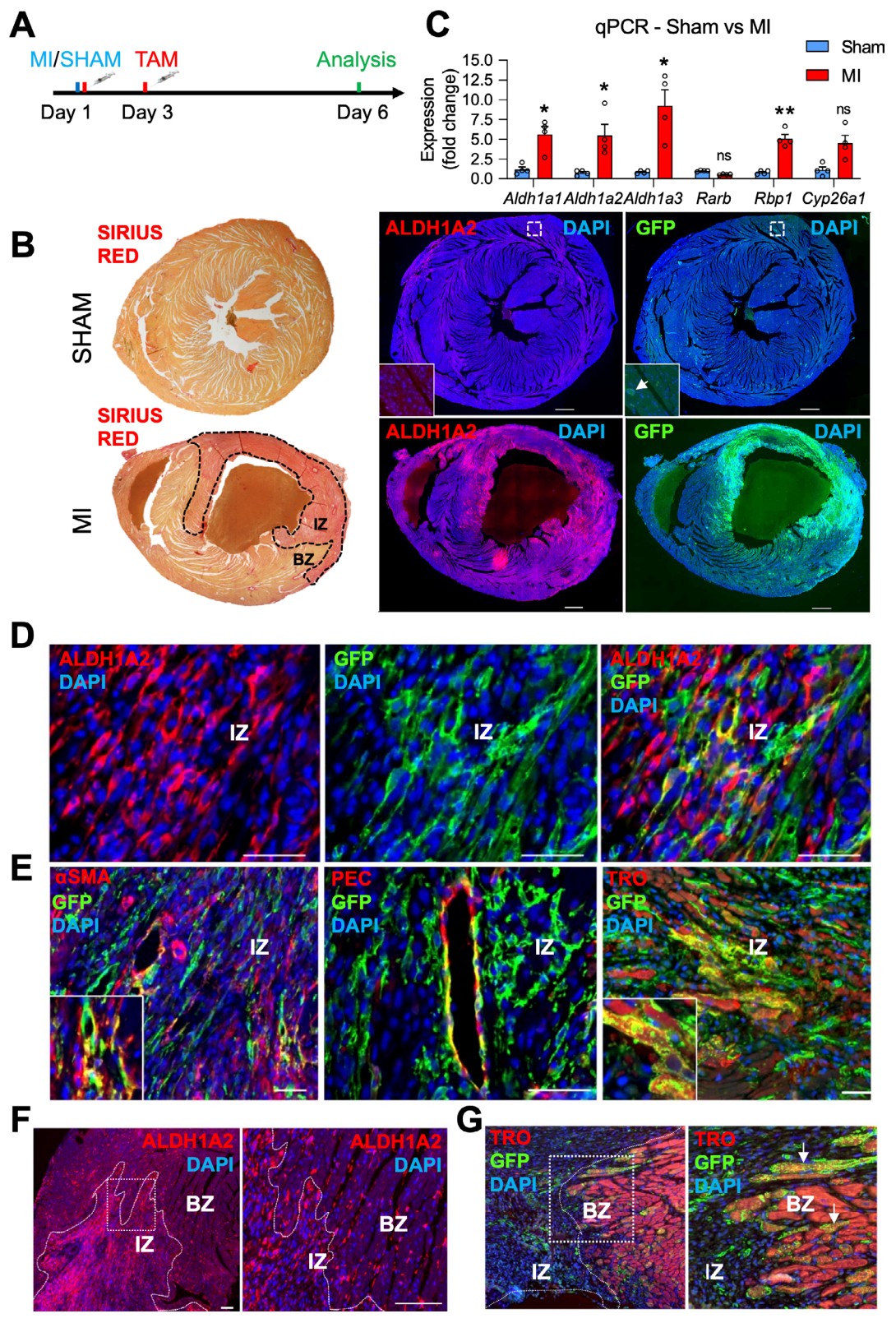

**Figure 4.** *The RARECreER^T2 line labels several cell-types, including cardiomyocytes, in adult hearts subjected to myocardial infarction.* (**A**) Schematic illustrating the lineage tracing experiments performed in RARECreER^T2; mTmG mice subjected to myocardial infarction (MI). Tamoxifen was administered twice (30 min and 48 hr after surgery) and hearts were analyzed 6 days post MI. (**B**) IF on infarct and sham hearts from RARECreER^T2 mice reveals ALDH1A2 and GFP are highly enriched in and around the infarct zone (IZ) (marked by Sirius Red staining, dotted black line) while minimal

*Figure 4 continued on next page*

Figure 4 continued

ALDH1A2 and GFP staining is detected in sham hearts. BZ = border zone of injury. (C) qPCR analysis on RNA extracted from infarct and sham hearts reveals *Aldh1a1,2* and *3* and *Rbp1* are upregulated after MI. The RA targets *Rarb* and *Cyp26a1* are not significantly altered. Data are expressed as fold change vs controls and columns are means ± SEM (n = 4 hearts). (D) Co-IF for ALDH1A2 and GFP demonstrates minimal co-staining in RARECreER[T2] MI hearts. (E) Co-IF for GFP plus αSMA, PECAM1 or Troponin T demonstrates an RA response in activated fibroblasts/smooth muscle cells, coronary vessels, and cardiomyocytes respectively in RARECreER[T2] MI hearts. (F) Closer analysis of RARECreER[T2] infarct hearts reveals ALDH1A2 protein localization to the infarct zone and border zone. GFP+ cells also localize to both regions, and many of the GFP cells in the border zone are cardiomyocytes as demonstrated by co-IF for Troponin T (white arrows). Data information: TRO = Troponin T, SMA = smooth muscle actin, PEC = PECAM1. All statistics two tailed t-test assuming unequal variance, *p < 0.05, **p < 0.01, ns = not significant. Scale bars: mosaics 100 µM, close ups 40 µM. p = 0.017 (C, *Aldh1a1*), p = 0.042 (C, *Aldh1a2*), p = 0.048 (C, *Aldh1a3*), p = 0.0033 (C, *Rbp1*), p = 0.11 (C, *Cyp26a1*).

(*Figure 5F* and *Figure 5—figure supplement 2C*). These patches were observed in 5 out of 10 RAKO mice and only 1 out of 11 control mice (*Figure 5—figure supplement 2D*, table). Moreover, RAKO mice exhibited higher mortality rates (Early death: RAKO 4 of 14 vs. CTL 0 of 11) before analysis at 6 days post MI (*Figure 5—figure supplement 2D*, table). Thus, dampening the RA response prior to MI leads to increased cardiomyocyte apoptosis and mortality, suggesting RA signaling plays a protective role in damaged heart muscle.

A previous study using exogenous RA supplementation in mice subjected to ischaemia/reperfusion determined that the RA pathway reduced apoptosis in damaged hearts by modulating the MAP kinase pathway. This was suggested to be through direct transcriptional regulation of *Adam10*, a gene which encodes for a protease that cleaves and inactivates the RAGE10 (Receptor for Advanced Glycation End products) receptor, a positive regulator of the ERK1/2 MAP kinase pathway (*Zhu et al., 2015*). However, we did not observe any differences in ERK1/2 activation, as determined by IF for phospho-ERK1/2, in our RAKO mice (*Figure 5—figure supplement 3A-B*). Additionally, treatment of primary cardiomyocytes with RA or BMS493 did not lead to significant changes in *Adam10* expression (*Figure 5—figure supplement 3C-D*). Taken together, these data suggest the anti-apoptotic effects of RA observed in our model of RA depletion to be unrelated to *Adam10*/MAP-Kinase signaling.

## RA treatment in embryonic cardiomyocytes promotes a notable transcriptional response and regulates genes involved in cardiac repair, including *Tgm2* and *Ace1*

The labeling of cardiomyocytes with our RARECreER[T2] line, as well as the increase in apoptosis observed in *Aldh1a*-null mice after MI led us to hypothesize that RA signaling plays a protective role specifically in cardiomyocytes. To decipher the underlying mechanisms, we isolated primary cardiomyocytes from E18.5 hearts and treated them with 100 nM RA for 48 hr to mimic long-term exposure to RA, which is normally experienced after MI. We then extracted RNA and used the samples for high-throughput sequencing (*Figure 6A*, schematic). Analysis of the sequencing data identified several canonical RA targets such as *Rarb*, *Cyp26b1* and *Cyp26a1* to be top hits among all genes analyzed (*Figure 6B*, *Supplementary file 1*), and gene ontology analysis revealed enrichment of GO terms for inflammatory and DNA damage responses (*Figure 6C* and *Figure 6—figure supplement 1*). Furthermore, several genes previously shown to be important in cardiac development and MI were also determined to be either up or downregulated (*Figure 6B*). Of note, *Tgm2*, which has been shown to promote ATP synthesis and thus limit damage after MI, was significantly upregulated (*Figure 6B*; *Szondy et al., 2006*). We also noticed that RA treatment significantly repressed the expression of Angiotensin converting enzyme 1 (*Ace1*) (*Figure 6B*). ACE1 is responsible for converting angiotensin 1 into angiotensin 2, which in turn activates the renin angiotensin system and subsequent vasoconstriction. Upregulation of ACE1 has been observed in rodent hearts after MI and is generally considered to be harmful (*Sun, 2010*). Indeed, ACE inhibitors are commonly used to treat patients recovering from MI, and they have shown great success in various clinical trials (*Flather et al., 2000*). Moreover, a connection between RA signaling, the RAS system, and cardiomyocyte-specific apoptosis has been previously established in vitro (*Palm-Leis et al., 2004*; *Guleria et al., 2011*). Respective activation and repression of *Tgm2* and *Ace1* after 48 hr RA treatment was confirmed by qPCR analysis (*Figure 6D*). *Tgm2* and *Ace1* modulation by RA signaling was also observed with 9-hr RA-treatment (*Figure 6E*). Next, we treated cardiomyocytes with BMS493 for 48 hr, and observed that *Tgm2* expression was repressed while *Ace1* levels were increased, consistent with our hypothesis that both genes are regulated by RA

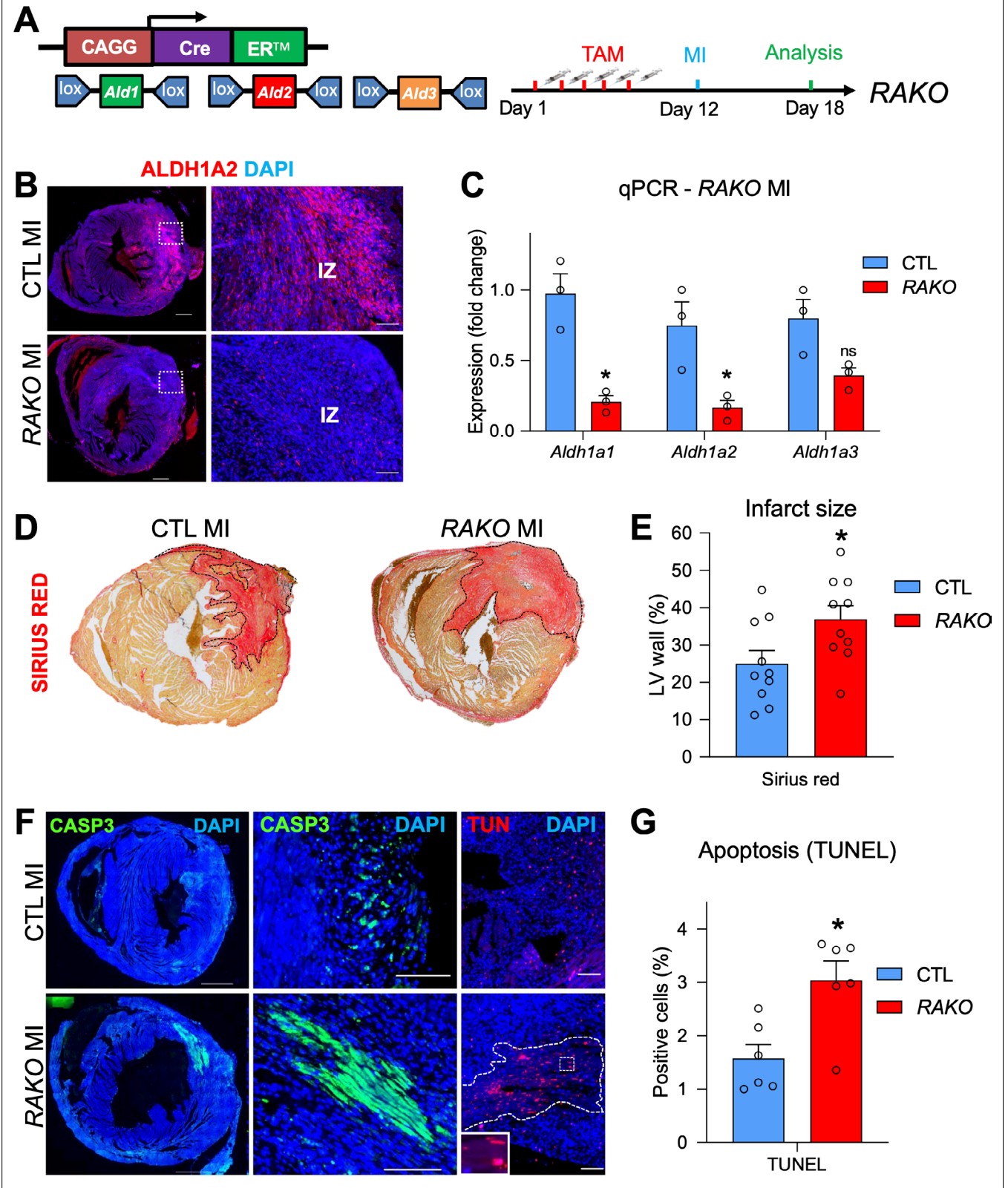

**Figure 5.** Depletion of RA signaling leads to larger infarct zones and increased apoptosis. (**A**) Schematic illustrating strategy used to delete floxed alleles of the *Aldh1a1,2,3 (Ald1,2,3)* enzymes with the CAGGCreER line (mutant mice referred to as RAKOs). Five daily doses of tamoxifen were administered 1 week prior to surgery and operated hearts were analyzed 6 days post MI. (**B**) IF analysis reveals a significant decrease of ALDH1A2 protein in RAKOs when compared to CAGGCreER negative (control (CTL)) hearts. (**C**) qPCR analysis of RNA extracted from infarct hearts reveals

*Figure 5 continued on next page*

*Figure 5 continued*

significant decreases in *Aldh1a1* and *Aldh1a2* expression in RAKOs when compared to controls. *Aldh1a3* expression is also reduced, though not significantly. Data are expressed as fold change vs. controls and columns are means ± SEM (n = 3 hearts). (**D**) Sirius red detection of collagen deposition demonstrates increased infarct size in RAKO hearts when compared to controls. See *Figure 5—figure supplement 2A* for representative images of all hearts analyzed. (**E**) Quantification of infarct size in RAKO and control hearts. The infarct areas were measured with ImageJ software and were normalized to the total area of the left ventricle. Columns are means ± SEM (n = 10 hearts). See also *Figure 5—source data 1*. (**F**) Active caspase three and TUNEL stainings reveal increased apoptosis in RAKO hearts when compared to controls. RAKO hearts have visible 'patches' of apoptotic cells (lower middle panel; white outline in lower right panel (magnified in inset)). (**G**) Quantification of TUNEL⁺ cells in RAKO and control infarct hearts using ImageJ software. Columns are means ± SEM (n = 6 hearts). See also *Figure 5—source data 2*. Data information: CASP3 = active caspase 3, TUN = TUNEL, IZ = infarct zone. All statistics two tailed t-test assuming unequal variance, *p < 0.05, ns = not significant. Scale bars: mosaics 100 μM, close ups 40 μM. p = 0.011 (C, *Aldh1a1*), p = 0.026 (C, *Aldh1a2*), p = 0.26 (C, *Aldh1a3*), p = 0.028 (E, RAKO), p = 0.010 (G, RAKO). See also *Figure 5—figure supplements 1–3*.

The online version of this article includes the following figure supplement(s) for figure 5:

**Source data 1.** Numerical source data for *Figure 5E*.

**Source data 2.** Numerical source data for *Figure 5G*.

**Figure supplement 1.** The CAGGCreER line is highly active in adult hearts but leads to incomplete excision of *Aldh1a* floxed alleles during MI, and *Aldh1a2* is efficiently deleted during late cardiac development.

**Figure supplement 2.** Increased infarct zones in RAKO hearts subjected to myocardial infarction.

**Figure supplement 3.** MAP kinase signaling is not significantly altered in RAKO infarcted hearts.

**Figure supplement 3—source data 1.** Numerical source data for *Figure 5—figure supplement 3B*.

signaling (*Figure 6F*). *Ace1* is highly expressed by endothelial cells in the heart, and it is possible the repression of *Ace1* by RA may be a result of contaminating endothelial cells. To address this issue, we removed endothelial cells from our cardiomyocyte cultures using CD31 magnetic beads, then treated enriched cardiomyocytes with RA for 48 hr (*Figure 6—figure supplement 2A-B*). Consistently, RA treatment repressed *Ace1* expression (*Figure 6G*). *Tgm2* and *Ace1* were also upregulated in MI hearts when compared to sham controls (*Figure 6—figure supplement 2C*). Finally, to see if *Tgm2* and *Ace1* are direct or indirect RA targets, we treated cultured cardiomyocytes with RA for only 3 hours and added the translational inhibitor cycloheximide to block non-specific activation of *Tgm2/Ace1*. No changes in *Tgm2* or *Ace1* expression were detected with the 3 hr RA treatment with or without cycloheximide, suggesting that *Tgm2* and *Ace1* are indirect RA targets. Importantly, *Rarb* expression was still activated by 3 hr RA treatment with or without cycloheximide (*Figure 6—figure supplement 2D*). Taken together, these data show that RA can stimulate a significant transcriptional response in primary cardiomyocytes, and that some of the identified genes, such as *Tgm2* and *Ace1*, which are likely indirect RA targets, are known to play important roles in cardiac repair.

## Discussion

Here, we have characterized a novel, tamoxifen-inducible reporter for RA signaling in mice that can be used to permanently label RA-responsive cells and trace their descendants in specific tissues and organs. By performing several inductions and analyzing embryos at different time-points during gestation, we show that our line faithfully recapitulates the pattern of RA activity observed with other well-characterized RA reporters. Notably, at embryonic stages, TAM-induced RARE-CreERᵀ² transgenics display specific labeling in the somites, limbs, forebrain, heart, and caudal regions of the body, consistent with well described roles for the RA pathway (*Duester, 2008*; *Niederreither and Dollé, 2008*; *Rhinn and Dollé, 2012*). We also characterize a novel and direct RA response in cardiomyocytes during multiple phases of embryonic development, as well as after MI.

An interesting finding of our study is the highly dynamic response of the RARECreERᵀ² line in various organs when activated at different time-points. Tamoxifen administration at E6.5 and E7.5 revealed very efficient labeling of multiple organs and tissues, while pulses given at later time-points such as E8.5 and E10.5 led to reduced recombination in most regions of the embryo. Overall, these data are consistent with previously described early roles of RA signaling in early embryonic germ layers and in body axis patterning, limb bud formation and neural tube patterning (*Stratford et al., 1999*; *Niederreither et al., 2002a*; *Niederreither et al., 1999*; *Diez del Corral et al., 2003*). Other organs that have continuous RA activity throughout development such as the retina, somites, forebrain, and heart

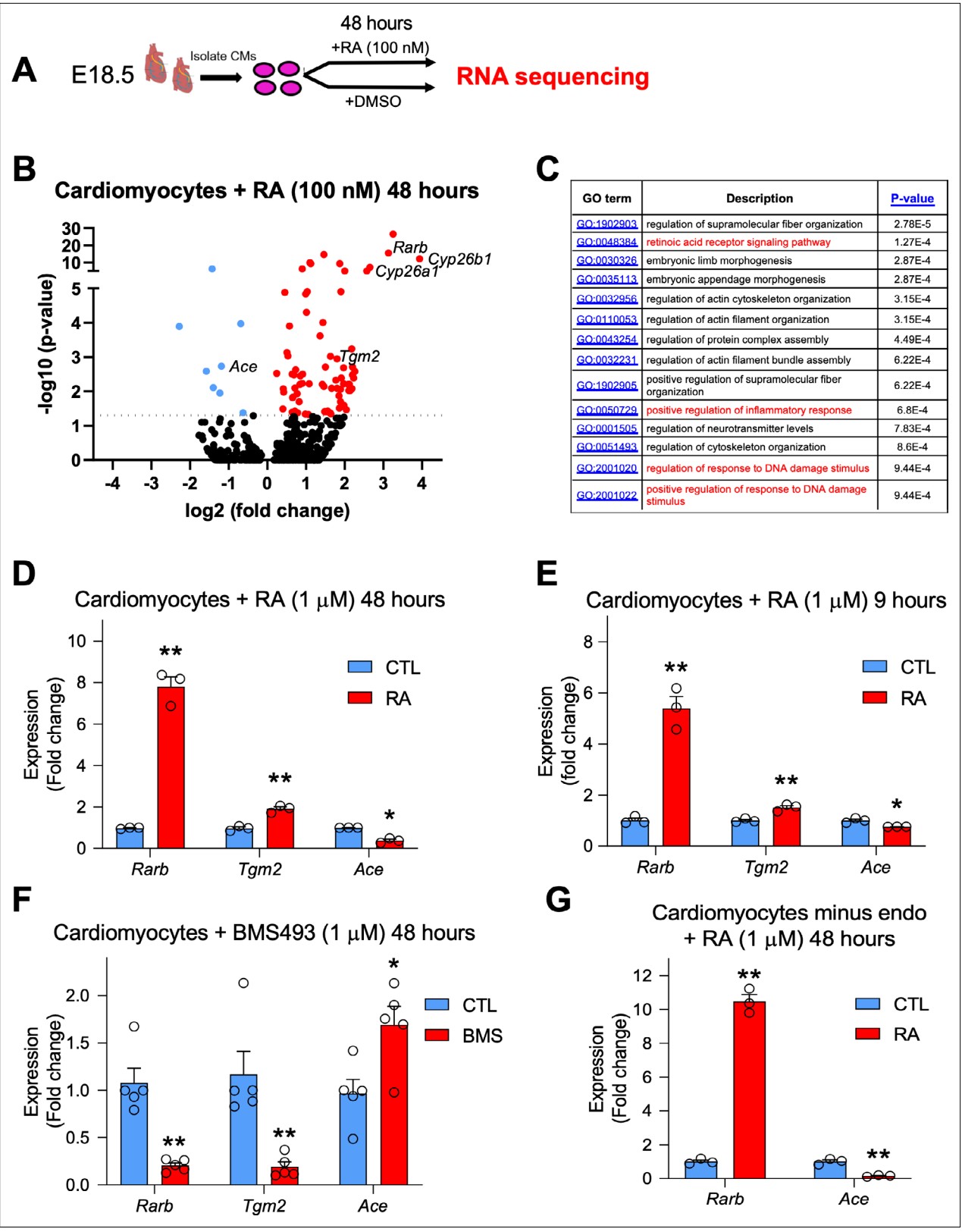

**Figure 6.** RA treatment in embryonic cardiomyocytes promotes a notable transcriptional response and regulates genes involved in cardiac repair such as *Tgm2* and *Ace1*. (**A**) Schematic illustrating the strategy used to isolate primary cardiomyocytes from E18.5 hearts. Cultured cardiomyocytes were treated for 48 hr with 100 nM RA. RNA was extracted, libraries were prepared with oligo (dT) primers and single end sequencing was performed on RA-treated and DMSO-treated cells (n = 4 biological replicates). (**B**) Volcano plot analysis of RNA sequencing results performed with Graphpad

*Figure 6 continued on next page*

*Figure 6 continued*

software. Dotted line represents significance threshold equivalent to p < 0.05. Several canonical RA targets (*Rarb; Cyp26a1; Cyp26b1*; red) are significantly upregulated. Genes involved in cardiac repair such as Transglutaminase 2 (*Tgm2*) (red) and Angiotensin converting enzyme 1 (*Ace1*) (blue) are also significantly altered. DESeq analysis of genome aligned reads was performed using proprietary Genomatix software. Only the top 500 genes were included in the volcano plot analysis. Blue dots represent downregulated genes. (**C**) Gene ontology (GO) analysis of biological processes using Gorilla software. Only the top 500 genes were included in the GO analysis. (**D**) qPCR analysis confirms upregulation of *Tgm2* and repression of *Ace1* mRNA levels in primary cardiomyoctes treated with 1 μM RA for 48 hr (n = 3 technical replicates). (**E**) Acute nine hour 1 μM RA treatment in primary cardiomyocytes promotes *Tgm2* upregulation and *Ace1* repression as shown by qPCR analysis (n = 3 technical replicates). (**F**) Treatment of primary cardiomyocytes with the RAR signaling reverse agonist BMS493 (1 μM) for 48 hr reveals a decrease in *Tgm2* expression and an increase in *Ace1* expression levels as shown by qPCR analysis (n = 5, experiment performed once in triplicate and once in duplicate). (**G**) Removal of endothelial cells (endo) with CD31-magnetic beads followed by 48 hr 1 μM RA treatment on purified primary cardiomyocytes reveals significant repression of *Ace1* expression as shown by qPCR analysis (n = 3 technical replicates). Data information: For all graphs data are expressed as fold change vs. controls and columns are means ± SEM. *Tgm2* = Transglutaminase 2, Ace = angiotensin converting enzyme. All statistics two tailed t-test assuming unequal variance, *p < 0.05, **p < 0.01. p = 0.0046 (D, *Rarb*), p = 0.0025 (D, *Tgm2*), p = 0.011 (D, *Ace*), p = 0.0093 (E, *Rarb*), p = 0.0093 (E, *Tgm2*), p = 0.043 (E, *Ace*), p = 0.0041 (F, *Rarb*), p = 0.015 (F, *Tgm2*), p = 0.0020 (F, *Ace*), p = 0.0012 (G, *Rarb*), p = 0.0091 (G, *Ace*). See also *Figure 6—figure supplements 1–2*.

The online version of this article includes the following figure supplement(s) for figure 6:

**Figure supplement 1.** Gene ontology analysis of RNA sequencing data.

**Figure supplement 2.** *Tgm2* and *Ace1* are indirect RA targets in cultured cardiomyocytes.

can be labeled using the RARECreER^T2 line with later pulses of tamoxifen (*Cunningham and Duester, 2015*; *Niederreither and Dollé, 2008*). The biological relevance of these results not only validate our RARECreER^T2 line; they also highlight its sensitivity and vast potential as a reliable tool for studying RA signaling and the fate of RA responding cells in vivo. A limitation with the RARECreER^T2 is that it relies exclusively on the RARE element from the *Rarb* promoter, meaning it only functions in cells where *Rarb* is activated by RA. Moreover, the RARECreER^T2 line may also fail to detect RA signaling activation that is independent of RAR receptors (*Schug et al., 2007*).

Previous studies have shown that RA signaling is involved in myocardial compaction but that RA does not act directly on cardiomyocytes (*Tran and Sucov, 1998*; *Stuckmann et al., 2003*). Instead, RA signaling was believed to act in an autocrine or extra cardiac manner in the epicardium and liver/placenta, respectively (*Brade et al., 2011*; *Shen et al., 2015*). Using our novel RARECreER^T2 line, we clearly demonstrate that cardiomyocytes do indeed respond to RA. Moreover, we have validated this response through in vivo and in vitro modulation of the RA pathway. However, it is still not clear whether this response plays an important role in myocardial compaction. Deletion of the RXRA receptor in the myocardium does not lead to an obvious heart phenotype (*Tran and Sucov, 1998*), but perhaps other receptors compensate for this loss of function. It is also possible that RA signaling may have essential roles in fine-tuning the patterning and maturation of the compact myocardium. One way of addressing this would be to perform a simultaneous myocardial-specific deletion of the three RARs. The deletion would have to target all three alleles since the RARs display high functional redundancy (*Mark et al., 2006*). Such a study would be useful in further delineating the roles of RA signaling during myocardial compaction and would provide additional insight into how RA regulates the proliferation, differentiation and/or maturation of cardiomyocytes.

The cardiomyocyte response as well as the dynamic pattern of ALDH1A2 protein localization observed during late stages of heart development is intriguing. Why do cardiomyocytes continue to respond to RA signaling at these time-points? Could RA be promoting the differentiation and maturation of these cells? Do the RA-responsive cells represent unique subpopulations with specific roles in the heart? All of these scenarios are plausible and should be addressed in future studies. The fact that ALDH1A2 protein is produced by alternative cell-types other than the epicardium is logical, since during later time-points, when the heart's compact myocardial layer is thickened, it is unlikely RA can diffuse easily to cardiomyocytes located in deeper portions of the ventricular wall. Through a series of co-IF and lineage-tracing experiments we suggest that the ALDH1A2^+ cells at these time-points are cardiac fibroblasts or other connective tissue cells derived from the epicardium. However, more detailed analyses such as FACS analysis with multiple markers or single-cell RNA sequencing would have to be carried out in order to confirm these findings. It would also be interesting to investigate if these same cell-types also produce ALDH1A2 after MI.

It is well known that RA signaling plays a protective role in the heart after acute damage. Treatment of rats with RA reduces cardiac hypertrophy and remodeling after MI and deficiency of RA leads to larger infarct zones (*Paiva et al., 2005*; *Minicucci et al., 2010*). In mice, RA treatment also leads to smaller infarct zones and reduced apoptosis after ischaemia reperfusion (*Zhu et al., 2015*). Our observation of increased scarring and apoptosis after infarct in RAKO mice is consistent with these studies. However, other studies suggest that the protective effects of RA signaling is context dependent, and that short-term localized exposure to retinoids after MI may even be harmful (*Danzl et al., 2019*). In healthy non-infarcted mice, deletion of the RARA receptor has detrimental effects, mainly due to increased reactive oxygen species and calcium mishandling, while long-term RA treatment leads to cardiac hypertrophy and overall negative effects (*Zhu et al., 2016*; *Silva et al., 2017*). Overall, these contradictory findings suggest that the effects of RA signaling in the heart are highly dependent on the dose and length of exposure to RA, meaning that for RA signaling to be protective, it needs to be tightly regulated.

With our RARECreER$^{T2}$ mice we have detected an RA response in several cell types in MI hearts. Moreover, we report upregulation of ALDH1A2 in the injured heart and propose it is the main driver of the RA response post MI. We also propose that, based on the proximity of the ALDH1A2$^+$ cells to the GFP$^+$ cells, the RA response in the infarct region of the heart is localized. However, reports have shown that *Aldh1a2* is also upregulated in the epicardium after MI (*Kikuchi et al., 2011*) and, given the length of our lineage tracing experiments (6 days), we cannot rule out epicardial-specific *Aldh1a2* as the main driver of RA signaling in injured hearts. Interestingly, cardiomyocytes were also responsive to RA signaling in MI hearts, especially in the border zone. However, the number of RA-responsive cardiomyocytes was relatively low. This may be explained by the RARECreER$^{T2}$ line not being sensitive enough to detect the full spectrum of the RA-response in the injured myocardium. The importance of myocardial RA signaling in protecting infarct hearts is supported by the increased cardiomyocyte apoptosis observed in RAKO mice.

To our knowledge, this is the first study to conduct RNA sequencing on primary cardiomyocytes treated with RA. Despite traditionally being considered non-responsive cells, RA treatment led to a notable transcriptional response in cardiomyocytes. In many ways, this validates our observations with the RARECreER$^{T2}$ line, in which cardiomyocytes, especially at later time-points are highly responsive to RA. However, these results need to be interpreted with caution since the cardiomyocytes used for the RNA sequencing were isolated from E18.5 hearts, whose response to RA may not accurately reflect the transcriptional landscape during initial phases of myocardial compaction (E10.5-E14.5) or after MI, in which the myocardium experiences severe hypoxia and inflammation. Furthermore, RA signaling can regulate gene expression post-transcriptionally and it is likely that many of the genes identified, such as *Tgm2* and *Ace1*, are indirect RA targets.

Among the many genes identified from the RNA sequencing analysis, *Tgm2* attracted our attention since it has previously been associated with regulating ATP synthesis after MI. In fact, *Tgm2* knockout mice exhibit larger infarcts when compared to controls, much like our RAKO mice (*Szondy et al., 2006*). More recently, however, *Tgm2* has been shown to promote cardiac fibrosis in MI hearts and pharmacological inhibition of TGM2 led to smaller infarcts (*Wang et al., 2018*). It is thereby possible that the effects of *Tgm2* on heart repair, as with RA signaling, are dependent on the level of activity.

Another important finding relates to RA repression of *Ace1*, an essential RAS activator (*Sun, 2010*). Previous work has identified numerous links between RA signaling and RAS but the mechanisms through which this crosstalk occurred were unclear since no RA targets from the RAS pathway had been identified (*Guleria et al., 2011*; *Palm-Leis et al., 2004*; *Takeda et al., 2000*; *Dechow et al., 2001*; *Haxsen et al., 2001*; *Zhong et al., 2004*; *Zhou et al., 2012*). The finding that *Ace1* expression is repressed by RA in cardiomyocytes demonstrates a direct connection between RA signaling and RAS. Moreover, it has important implications for cardiac disease, since *Ace1* upregulation after MI is long known to have negative effects on cardiac remodeling (*Sun, 2010*). Indeed, ACE inhibitors have consistently led to improved patient outcomes in clinical trials (*Flather et al., 2000*). Furthermore, one study has even shown that ACE1 can be detected in cardiomyocytes from human infarct hearts (*Hokimoto et al., 1996*). It is thus possible that combining RA, which specifically inhibits *Ace1* expression, with ACE inhibitors may have synergistic beneficial effects in protecting cardiomyocytes post-MI, an interesting prospect for future studies aimed at improving the survival of patients suffering from heart disease.

In summary, we have shown that cardiomyocytes are responsive to RA signaling and that depletion of the RA pathway leads to increased cardiomyocyte apoptosis after MI. These findings have important implications in the heart development and cardiac regeneration research fields.

# Materials and methods

## Key resources table

| Reagent type (species) or resource | Designation | Source or reference | Identifiers | Additional information |
|---|---|---|---|---|
| Gene (*Mus musculus*) | *Aldh1a1* | Mouse genome informatics | MGI: 1353450 | |
| Gene (*Mus musculus*) | *Aldh1a2* | Mouse genome informatics | MGI: 107928 | |
| Gene (*Mus musculus*) | *Aldh1a3* | Mouse genome informatics | MGI: 1861722 | |
| Gene (*Mus musculus*) | *Rarb* | Mouse genome informatics | MGI: 97857 | |
| Gene (*Mus musculus*) | *Ace1* | Mouse genome informatics | MGI: 87874 | |
| Gene (*Mus musculus*) | *Tgm2* | Mouse genome informatics | MGI: 98731 | |
| Strain, strain background (*Mus musculus*, male and female) | RARECreER$^{T2}$ | This paper | *Tg(RARE-Hspa1b-cre/ER$^{T2}$)* RRID:MGI:6726566 | Mixed genetic background |
| Strain, strain background (*Mus musculus*, male and female) | R26L | Soriano, 1999 | *Gt(ROSA)26Sor$^{tm1Sor}$* RRID:MGI:1861932 | Mixed genetic background |
| Strain, strain background (*Mus musculus*, male and female) | mTmG | Muzumdar et al., 2007 | *Gt(ROSA)26Sor$^{tm4(ACTB-tdTomato,-EGFP)Luo}$* RRID:MGI:3716464 | Mixed genetic background |
| Strain, strain background (*Mus musculus*, male and female) | CAGGCreER | Hayashi and McMahon, 2002 | *Tg(CAG-cre/Esr1*)5Amc* RRID:MGI:2182767 | Mixed genetic background |
| Strain, strain background (*Mus musculus*, male and female) | Aldh1a1$^{fl}$ | Matt et al., 2005 | | Mixed genetic background |
| Strain, strain background (*Mus musculus*, male and female) | Aldh1a2$^{fl}$ | Vermot et al., 2006 | | Mixed genetic background |
| Strain, strain background (*Mus musculus*, male and female) | Aldh1a3$^{fl}$ | Dupé et al., 2003 | | Mixed genetic background |
| Strain, strain background (*Mus musculus*, male and female) | WT1CreER$^{T2}$ | Zhou et al., 2008 | *Wt1$^{tm2(cre/ERT2)Wtp}$* RRID:MGI:3801682 | Mixed genetic background |
| Biological sample (*Mus musculus*) | Primary cardiomyocytes | This paper | N/A | Freshly isolated from E18.5 embryonic hearts |
| Antibody | Anti-GFP (chicken polyclonal) | Abcam | Cat#: AB13970 RRID:AB_300798 | IF (1:400) |
| Antibody | Anti-ALDH1A2 (rabbit polyclonal) | Sigma Aldrich | Cat#: ABN420 | IF (1:300) |
| Antibody | Anti-Myosin D (mouse monoclonal) | Santa Cruz | Cat#: sc32758 RRID:AB_627978 | IF (1:200) |
| Antibody | Anti-MF20 (mouse monoclonal) | DSHB | Cat#: AB_2147781 RRID:AB_2147781 | IF (1:20) |
| Antibody | Anti-αSMA (mouse monoclonal) | Santa Cruz | Cat#: sc53015 RRID:AB_628683 | IF (1:1000) |
| Antibody | Anti-WT1 (mouse monoclonal) | Agilent | Cat#: M3561 RRID:AB_2304486 | IF (1:100) |
| Antibody | Anti-Vimentin (chicken polyclonal) | Abcam | Cat#: AB24525 RRID:AB_778824 | IF (1:1000) |
| Antibody | Anti-active caspase 3 (rabbit polyclonal) | R & D | Cat#: AF835 RRID:AB_2243952 | IF (1:1000) |
| Antibody | Anti-active caspase 3 (rabbit polyclonal) | R & D | Cat#: AF835 RRID:AB_2243952 | IF (1:200) |
| Antibody | Anti-phospho-ERK (rabbit monoclonal) | Cell signalling technology | Cat#: 4370 S RRID:AB_2315112 | F (1:400) |
| Antibody | Anti-Troponin T (mouse monoclonal) | Invitrogen | Cat#: MA5-12960 RRID:AB_11000742 | IF (1:300) |
| Antibody | Anti-GATA4 (goat polyclonal) | Santa Cruz | Cat#: sc1237 RRID:AB_2108747 | IF (1:500) |

*Continued on next page*

*Continued*

| Reagent type (species) or resource | Designation | Source or reference | Identifiers | Additional information |
|---|---|---|---|---|
| Antibody | Anti-PECAM-1 (goat polyclonal) | Santa Cruz | Cat#: sc1506 RRID:AB_2161037 | IF (1:200) |
| Antibody | Anti-SM22a (rabbit polyclonal) | Abcam | Cat#: AB14106 RRID:AB_443021 | IF (1:400) |
| Antibody | Anti-ALDH1A3 (rabbit polyclonal) | Sigma Aldrich | Cat#: HPA046271 RRID:AB_10965992 | IF (1:200) |
| Antibody | Anti-ALDH1A1 (rabbit monoclonal) | Abcam | Cat#: ab52492, RRID:AB_867566 | IF (1:200) |
| Antibody | Anti-Mouse IgG Cy3 (donkey polyclonal) | Jackson immunoreseach | Cat#: 715-165-150; RRID:AB_2340813 | IF (1:400) |
| Antibody | Anti-Mouse IgG, Alexa Fluor 647 (donkey polyclonal) | Jackson immunoreseach | Cat#: 715-605-151; RRID:AB_2340863 | IF (1:400) |
| Antibody | Anti-Mouse IgG, Alexa Fluor 488 (donkey polyclonal) | Jackson immunoreseach | Cat#: 715-545-150; RRID:AB_2340846 | IF (1:400) |
| Antibody | Anti-Rabbit IgG Cy3 (donkey polyclonal) | Jackson immunoreseach | Cat#: 711-165-152; RRID:AB_2307443 | IF (1:400) |
| Antibody | Anti-Rabbit IgG, Alexa Fluor 647 (donkey polyclonal) | Jackson immunoreseach | Cat#: 711-605-152; RRID:AB_2492288 | IF (1:400) |
| Antibody | Anti-Goat IgG, Alexa Fluor 647 (donkey polyclonal) | Jackson immunoreseach | Cat#: 705-605-147; RRID:AB_2340437 | IF (1:400) |
| Antibody | Anti-Chicken IgG, Alexa Fluor 488 (donkey polyclonal) | Jackson immunoreseach | Cat#: 703-545-155; RRID:AB_2340375 | IF (1:400) |
| Sequence-based reagent | Rarb_F | This paper | qPCR primer | GTCAGCGCTGGAATTCGT |
| Sequence-based reagent | Rarb_R | This paper | qPCR primer | CACCGGCATACTGCTCAA |
| Sequence-based reagent | Rbp1_F | This paper | qPCR primer | TCTCCCTTCTGCACACACTG |
| Sequence-based reagent | Rbp1_R | This paper | qPCR primer | GCCATTGGCCTTCACACT |
| Sequence-based reagent | Cyp26a1_F | This paper | qPCR primer | GGAGCTCTGTTGACGATTGTT |
| Sequence-based reagent | Cyp26a1_R | This paper | qPCR primer | CCGGCTTCAGGCTACAGA |
| Sequence-based reagent | Raldh1_F | This paper | qPCR primer | CATCTTGAATCCACCGAAGG |
| Sequence-based reagent | Raldh1_R | This paper | qPCR primer | GCCATCACTGTGTCATCTGC |
| Sequence-based reagent | Raldh2_F | This paper | qPCR primer | GGCAGGATATTGACGACTCC |
| Sequence-based reagent | Raldh2_R | This paper | qPCR primer | TGAGCAGACACCGCTCAGT |
| Sequence-based reagent | Raldh3_F | This paper | qPCR primer | AGTCGGTGCTATTCGCTCTC |
| Sequence-based reagent | Raldh3_R | This paper | qPCR primer | TGAGGATTGCCAAAGAGGA |
| Sequence-based reagent | Cyp26b1_F | This paper | qPCR primer | CACTTTGCCCAGGAGGAAT |
| Sequence-based reagent | Cyp26b1_R | This paper | qPCR primer | CAGAAGGAAGTCTGGGCTTG |
| Sequence-based reagent | Ace1_F | This paper | qPCR primer | TGCAGCTCCTGGTACAGTTTT |
| Sequence-based reagent | Ace1_R | This paper | qPCR primer | AAGATTGCCAAGCTCAATGG |
| Sequence-based reagent | Adam10_F | This paper | qPCR primer | CTCAGGACCACTACTAGCAGCA |
| Sequence-based reagent | Adam10_R | This paper | qPCR primer | CCGTTTTTGAAAGGATGAGG |
| Sequence-based reagent | Tgm2_F | This paper | qPCR primer | GGTTTTGCTTGGGTTCTCC |
| Sequence-based reagent | Tgm2_R | This paper | qPCR primer | ACCTGCTGGCTGAGAGAGAT |
| Commercial assay or kit | In situ cell death detection kit, TMR red | Roche | Cat#: 12156792910 | |
| Commercial assay or kit | Neonatal cardiomyocyte isolation kit, mouse | Miltenyi | Cat#: 130-100-825 | |
| Commercial assay or kit | Nucleospin RNA, Mini kit for RNA purification | Machery Nagel | Cat#: 740955.250 | |

*Continued on next page*

*Continued*

| Reagent type (species) or resource | Designation | Source or reference | Identifiers | Additional information |
|---|---|---|---|---|
| Chemical compound, drug | Tamoxifen | Sigma Aldrich | Cat#: T5648 | |
| Chemical compound, drug | 4-Hydroxytamoxifen | Sigma Aldrich | Cat#: H6728 | |
| Chemical compound, drug | Corn oil | Sigma Aldrich | Cat#: C8267 | |
| Chemical compound, drug | BMS493 | TOCRIS | Cat#: 3509 | |
| Chemical compound, drug | Retinoic acid | Sigma Aldrich | Cat#: R2625 | |
| Chemical compound, drug | Antigen Unmasking Solution, Citrate-Based | Vector laboratories | Cat#: H-3300–250 | |
| Chemical compound, drug | Direct red 80 | Sigma Aldrich | Cat#: 365,548 | |
| Chemical compound, drug | Picric acid solution | VWR International | Cat#: 87897.18 | |
| Chemical compound, drug | Trypsin from porcine pancreas | Sigma Aldrich | Cat#: T4799 | |
| Chemical compound, drug | TRIzol | Invitrogen | Cat#: 15596026 | |
| Chemical compound, drug | LightCycler 480 SYBR Green I Master | Roche | Cat#: 04707516001 | |
| Chemical compound, drug | Buprenorphrine | Axience | Cat#: 03760087151244 | |
| Chemical compound, drug | Isofluorane | Med'Vet | Cat#:0890402663435 | |
| Chemical compound, drug | Gelatin | VWR | Cat#: 24350.262 | |
| Chemical compound, drug | Thimerosal | Sigma Aldrich | Cat#: T8784 | |
| Chemical compound, drug | Low melting agarose | Invitrogen | Cat#: 16520050 | |
| Software, algorithm | Fiji (ImageJ) | https://fiji.sc/ | RRID:SCR_002285 | |
| Software, algorithm | GraphPad | Prism | RRID:SCR_002798 | |
| Software, algorithm | Adobe Photoshop | Photoshop | RRID:SCR_014199 | |
| Software, algorithm | LightCycler 480 software | Roche | RRID:SCR_012155 | |

## Experimental design

The aim of the study was to investigate the role of endogenous Retinoic Acid (RA) signaling in the heart using a novel transgenic mouse RA-reporter (RARECreER[T2]) and an in vivo genetic approach of deletion of *Aldh1a1, Aldh1a2,* and *Aldh1a3* genes (i.e. mice lacking all sources of RA in vivo, an approach validated in *Chassot et al., 2020*) to challenge the dogma that cardiomyocytes, the principle cell type of the heart, are not responsive to RA signaling both during embryonic development and after MI. The number of samples was determined on the basis of experimental approach, availability, and feasibility required to obtain definitive results. The numbers of replicates are specified in the appropriated Materials and Methods sections. The researchers were not blinded during data collection or analysis.

## Mice

All animal work was conducted according to national and international guidelines and was approved by the local ethics committee (PEA-NCE/2013/88). The R26L, mTmG, WT1CreER[T2], *Aldh1a1*[fl], *Aldh1a2*[fl], *Aldh1a3*[fl], and CAGGCreER lines have been described previously (*Soriano, 1999*; *Muzumdar et al., 2007*; *Zhou et al., 2008*; *Matt et al., 2005*; *Vermot et al., 2006*; *Dupé et al., 2003*; *Hayashi and McMahon, 2002*). For embryonic RARECreER[T2] lineage tracing Cre activation was obtained by administration (gavage) of 200 mg/kg tamoxifen (Sigma-Aldrich) dissolved in corn oil (Sigma-Aldrich) to pregnant females at the indicated time-points. For adult myocardial infarction experiments (RARECreER[T2] lineage tracing and *Aldh1a1/a2/a3* deletion experiments), Cre activation was obtained by administration (intraperitoneal injection) of 100 mg/kg tamoxifen at the indicated time-points. Details of RA and BMS493 in vivo treatments are provided in the figure legends.

For generation of RARECreER[T2] mice, fertilized zygotes were obtained from super-ovulated B6D2F1 females mated to B6D2F1 males. Linearized DNA consisting of three copies of a 34 bp oligo (*Rossant*

*et al., 1991*) containing the direct repeat 5 (DR5) RARE of the murine *Rarb2* promoter upstream of the hsp68 promoter and *CreER^T2* gene was injected into the pronuclei of zygotes. Zygotes were then transferred into the oviducts of pseudo pregnant mice. All mice used in these experiments were heterozygous for the transgene and were back-crossed for more than five generations. Two founder lines were established yielding the expected RA-responses during embryonic development, and the line with the higher activity was used for further study.

## Myocardial infarction surgeries

Transgenic males 8–10 week-old were subjected to permanent ligation of left-anterior descending coronary artery. One dose of buprenorphine (0.1 mg/kg) was administrated subcutaneously for pre-operative analgesia. After 30 min, they were anesthetized by inhalation of 4 % isoflurane in a filled chamber and immediately intubated for artificial ventilation to maintain a respiratory rate of 135 /min with pure oxygen mixed with 1–2% isoflurane. A left-sided thoracotomy was performed between the third and fourth ribs: the pericardium was cut open and ligation of the descending branch of the left coronary artery was made 2.5 mm under the tip of the left auricle using 8–0 silk suture. Sham operated mice underwent the same procedure, except no ligation around the left coronary artery was made. Subsequently, the intercostal space, muscles of the external thoracic wall and skin were sutured with 6/0 polyester. All animals received 0.5 ml saline IP post-surgery to compensate for fluid loss and 0.1 mg/kg of buprenorphrine subcutaneously for analgesia.

## Whole-mount X-gal staining

For whole-mount X-gal analysis, embryos or dissected organs were fixed for 45 min in 0.1 % glutaraldehyde diluted in PBS, washed three times with wash buffer (2 mM $MgCl_2$, 0.2% NP-40, 0.1 % sodium deoxycholate in sodium phosphate buffer) and then incubated overnight (O/N) at 37 °C in X-gal staining solution (washing buffer +5 mM potassium ferrocyanide, 5 mM potassium ferricyanide and 1 mg/ml X-Gal substrate). For analysis on sections, samples were fixed O/N in 4 % paraformaldehyde, embedded in paraffin, cut at 5 µM on a microtome, de-paraffinized and counterstained with eosin.

## Whole-mount GFP staining

Embryos were collected on their respective days and fixed with 4 % paraformaldehyde overnight at 4 °C. Embryos were dehydrated using an ascending concentration of methanol prepared in PBS, and bleached with 6 % hydrogen peroxide O/N at 4 °C. On the following day, embryos were rehydrated using a descending concentration of methanol. Embryos were quenched with 0.3 M Glycine with Triton X (prepared in PBS) for 4–6 hr at room temperature. The Triton X concentrations were decided based on the stage of the embryos (0.5 % Triton X concentration was used for E10.5 embryos and 1.0 % Triton X concentration was used for E12.5 embryos). Samples were blocked in PBSGT (0.2 % Gelatin (VWR)), Triton X and 0.1 g/L of Thimerosal ((SIGMA), prepared in PBS) for 2 days at room temperature. Chicken Anti-GFP antibody and goat Anti-GATA4 were diluted in PBSGT at 1:500 concentrations. Samples were incubated with primary antibody solutions, 24 hr for E10.5 embryos and 1 week for E12.5 embryos at room temperature. Samples were washed six or more times over a day with PBSGT. Secondary antibodies were diluted in PBSGT at 1:500 concentration. Embryos were incubated in secondary antibodies for one (E10.5) to three (E12.5) day(s) at room temperature. Embryos were washed six or more times over a day with PBSGT.

## Image acquisition for whole-mount GFP staining

The protocol was adjected from a previously described protocol (*Hokimoto et al., 1996*). Embryos were embedded in 1.0 % low melting agarose (Invitrogen) prepared in PBS. The embedded embryos were transferred into amber color glass vials and dehydrated with ascending concentrations of methanol. BABB solution was prepared by mixing one portion of benzyl alcohol (SIGMA) with two portions of benzyl benzoate (SIGMA). Samples were treated with 50 % BABB diluted in methanol for 3–4 hr. Samples were then treated with the BABB solution until they were settled at the bottom of the vial. Samples were stored and mounted for imaging in BABB solution at room temperature.

Images were acquired using Zeiss LSM780 microscopy (or homemade light-sheet microscopy) at the PRISM Imaging Facility. 3D reconstructions and analysis were performed using Imaris software.

## Immunofluorescence, histological analysis, and sirius red staining

For IF experiments, tissues were fixed overnight in 4 % paraformaldehyde, progressively dehydrated and embedded in paraffin. 5 µM thick sections were rehydrated, boiled in a pressure cooker for 2 min with Antigen Unmasking Solution (Vector laboratories) and blocked in PBS solution containing 10 % normal donkey serum and 3 % BSA. All antibodies were applied overnight at 4 °C at the concentrations listed in the antibody table (see *Supplementary file 1*). Secondary antibodies were diluted 1:400 and applied at room temperature for 1 hr. TUNEL stainings were performed as IF experiments with the TMRRed In situ dell death detection kit (Roche). Imaging was performed with a motorized Axio Imager Z1 microscope (Zeiss) coupled with an AxioCam MRm camera (Zeiss), and images were processed with AxioVision LE and ImageJ.

For histological analysis, 5 -µM-thick sections were stained with haematoxylin and eosin according to standard procedures. For Sirius red staining, 5 -µM-thick sections were stained in Sirius red solution (Sirius red powder (SIGMA) dissolved in picric acid) for 1 hr at room temperature. Samples were then washed in acidified water before being dehydrated with three 100 % ethanol baths.

## Isolation of primary cardiomyocytes and treatment with RA and BMS493

E18.5 hearts were dissected, minced, and then digested in DMEM+ Trypsin (100 mg/ml) for three times 15 min at 37 °C with shaking. After each 15 -min incubation, the supernatant was removed and 5 ml FBS was added to stop the reaction. After the digestion, all solutions were pooled, run through a 70 µM filter and then spun down for 5 minutes at 1600 rpm at 4 °C. The supernatant was then removed, and the pellet resuspended in DMEM +8 % FBS. The cells were then plated for 1–2 hr on uncoated plastic wells of six-well plates for fibroblasts to adhere. The non-adherent cells containing the cardiomyocytes were then resuspended and plated on collagen coated (50 µg/ml) wells of six-well plates. The next day the media was changed and the cardiomyocytes grown to 50–60% confluency before being treated. For RA (SIGMA) and BMS493 (TOCRIS) treatments, solutions were pre-diluted in 100 % ethanol before being added directly to the media at final concentrations of 100 nM (RA for RNA-seq) or 1 µM (BMS493 and RA for qPCR analysis). For RARECreER$^{T2}$ activation in cultured cardiomyocytes, 4-Hydroxytamoxifen (4 µM) was supplemented together with RA, as indicated in the figure legends. For IF experiments, cells were fixed in PFA for 10 min on ice and antibody incubations were performed as described above.

For live cell imaging and RA dose titration experiments, cardiomyocytes were isolated from RARECreER$^{T2}$; mTmG P1-P5 hearts with the Neonatal Cardiomyocyte Isolation kit, mouse (Miltenyi 130-042-401) according to the manufacturer's protocol. All cells isolated were counted as cardiomyocytes and were visualized and quantified via Tomato (not activated by RA) or GFP (activated by RA) (both from mTmG transgene) fluorescence. To avoid bias from FBS, cells were washed twice with PBS and media was replaced with KnockOut DMEM (Gibco 10829018) supplemented with 10 % KnockOut serum replacement (Gibco 10828028), 1 % MEM non-essential amino acids (Gibco 11140050) and 1 % GlutaMAX prior to RA treatments. Live cell imaging was performed with AxioObserver - Zeiss (2011) and images were acquired for a total of 48 hr. For dose titration experiments, cells were fixed in methanol, stained with anti-GFP antibody (Abcam: a13970), and revealed with anti-chicken Cy5 antibody (Jackson: 03-175-155). Images were taken on a Zeiss LSM780 microscope.

## RT-qPCR

RNA was extracted from the bottom halves of infarcted or sham adult hearts and primary cardiomyocytes using TRIzol reagent (Invitrogen), following the manufacturer's instructions. For 3 hr RA treatment of cardiomyocyte experiments, the Nucleospin RNA kit (Machery Nagel) was used for RNA isolation. Reverse transcription was performed using the M-MLV reverse transcriptase in combination with oligo (dT) primers (Invitrogen) or random primers (Invitrogen) (3 hr RA treatment of cardiomyocytes). The cDNA was used as a template for quantitative PCR analysis using the SybrGREEN Master Kit (Roche) or the Universal probeLibrary system (3 hr RA treatment of cardiomyocytes) and a Light Cycler 1.5 (Roche). Primers used for the analysis are shown in *Supplementary file 1*. The expression levels were normalized to *Gapdh*. For each litter or experiment, ddCt values were normalized to one control dCt rather than the mean of control delta Cts. Primers (see primer table) were designed on the Universal Probe Library website (Roche).

## RNA sequencing analysis

TRIzol RNA extraction was performed on four biological replicates of primary E18.5 cardiomyocytes treated with 100 nM atRA or DMSO as a control. Libraries were prepared on a Beckman Fxp Automation system, using the Illumina TruSeq stranded polyA chemistry kit. For each sample 500 ng was used as input for library preparation. Single end sequencing with an average of 20 million reads per sample was performed with the Illumina HiSeq 2000 at the EMBL sequencing center (Heidelberg, Germany). Sequences were aligned to the mouse mm10 reference genome with Burrows-Wheeler Aligner (bwa) version 0.7.12-r104 using standard parameters. Differential analysis of gene expression was calculated with the DESeq2 program from proprietary Genomatix software with a cutoff value of $p < 0.05$ (see *Supplementary file 1*). The raw data files have been submitted to GEO database (GSE161429).

## Quantification of collagen and IF stainings on sections

For collagen staining quantification, the infarct areas from six sections per heart stained with Sirius Red were measured with ImageJ software and divided by the total area of the left ventricular wall. For phospho-ERK1/2 staining levels, the positive areas for four to five sections per heart were measured with ImageJ software and divided by the area of the left ventricular wall. For TUNEL quantification, the number of positive cells were calculated for four sections per heart with ImageJ software, and then divided by the total number of cells (DAPI⁺) in the left ventricular wall. For all analyses, sections were on average spaced by 40 µM, covering a total area of at least 250–400 µM.

## Statistical analyses

Sample sizes (individual embryos, litter numbers, and wells (in vitro experiments)) are reported in each figure legend. All cell counts were performed in standardized microscopic fields using either the ImageJ cell counter plug in or user-defined macros. All statistical analyses were conducted using Graphpad Prism. Data normality was tested by Shapiro-Wilk normality test and variances between groups were tested using F-test. Means between two groups were compared using two-tailed unpaired student's $t$-test and means between multiple groups were compared using two-way analysis of variance (ANOVA) followed by Tukey's multiple comparison tests. Statistical outliers were calculated using the Grubb's test. Results are displayed as arithmetic mean ± standard error of mean (SEM). Where indicated results are shown as fold change vs. controls. Statistically significant data are indicated as: * $p < 0.05$, **$p < 0.01$, and ***$p < 0.001$. Non-significant data is indicated as ns.

## Acknowledgements

We thank the staff of the animal facility and PRISM imaging platform at the iBV for their help.

## Additional information

### Funding

| Funder | Grant reference number | Author |
|---|---|---|
| Fondation de France | 00056856 | Andreas Schedl |
| Fondation ARC pour la Recherche sur le Cancer | SL22020605297 | Andreas Schedl |
| Agence Nationale de la Recherche | ANR-11-LABX-0028-01 | Fabio Da Silva Andreas Schedl |
| Ligue Contre le Cancer | equipe labelisée 2018 | Andreas Schedl |

The funders had no role in study design, data collection and interpretation, or the decision to submit the work for publication.

### Author contributions

Fabio Da Silva, Conceptualization, Data curation, Investigation, Methodology, Visualization, Writing – original draft, Writing – review and editing; Fariba Jian Motamedi, Investigation, Methodology,

Writing – original draft; Lahiru Chamara Weerasinghe Arachchige, Investigation, Methodology, Visualization; Amelie Tison, Investigation, Methodology; Stephen T Bradford, Jonathan Lefebvre, Investigation, Writing – original draft; Pascal Dolle, Norbert B Ghyselinck, Resources, Writing – original draft; Kay D Wagner, Conceptualization, Methodology, Supervision, Writing – original draft, Writing – review and editing; Andreas Schedl, Conceptualization, Funding acquisition, Project administration, Supervision, Writing – original draft, Writing – review and editing

## Author ORCIDs
Fabio Da Silva ⬤ http://orcid.org/0000-0002-8983-2238
Lahiru Chamara Weerasinghe Arachchige ⬤ http://orcid.org/0000-0003-4492-8946
Stephen T Bradford ⬤ http://orcid.org/0000-0002-9508-3894
Andreas Schedl ⬤ http://orcid.org/0000-0001-9380-7396

## Ethics
All animal work was conducted according to national and international guidelines and was approved by the local ethics committee (PEA-NCE/2013/88).

## Decision letter and Author response
Decision letter https://doi.org/10.7554/eLife.68280.sa1
Author response https://doi.org/10.7554/eLife.68280.sa2

---

# Additional files

## Supplementary files
• Supplementary file 1. Complete list of genes from RNA sequencing analysis on primary cardiomyocytes treated with RA. RNA sequencing data has been uploaded to the public functional genomics data repository gene expression omnibus (GSE161429).

• Transparent reporting form

## Data availability
RNA sequencing data have been deposited in GEO under accession code GSE161429.

The following dataset was generated:

| Author(s) | Year | Dataset title | Dataset URL | Database and Identifier |
|---|---|---|---|---|
| Da Silva F, Schedl A | 2021 | Genome wide sequencing analysis of primary cardiomyocytes treated with all-trans retinoic acid | https://www.ncbi.nlm.nih.gov/geo/query/acc.cgi?acc=GSE161429 | NCBI Gene Expression Omnibus, GSE161429 |

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
