## [Decision Letter]

[Editors' note: this paper was reviewed by Review Commons.]

**Acceptance summary:**

Your study provides a new mouse model to examine RA signaling in the heart and other tissues. Given the important role RA signaling has in development and tissue regeneration, we feel that your studies advance both technical and conceptual ideas in these fields.

---

## [Author Response]

Reviewer #1 (Evidence, reproducibility and clarity (Required)):SummaryThe manuscript entitled "Direct activation of RA signaling in cardiomyocytes protects hearts from apoptosis after myocardial infarction in mice" by Da Silva and collaborators describes a novel mechanism by which retinoic acid (RA), a small molecule known to be essential for heart development and function, exerts its role via a direct effect on cardiomyocytes. A new inducible RARE-Cre-ERT2 mouse model was developed where RA can be deleted at specific times of development and in multiple tissues. After validating that the new line mimics endogenous RA signaling by crossing RARE-Cre-ERT2 with R26L and mTmG lines, results show that cardiomyocytes from E11-E18 embryos and from adult mice subjected to myocardial infarction (MI) display RA activity. Elegant experiments were performed in embryos of pregnant females injected with RA. Additional data from mice subjected to MI document increased RA signaling in a variety of cells including cardiomyocytes. Mice with a 50% reduction of ALDH1A have larger infarcts and increased fibrosis and apoptosis. Mechanistically, RNA sequencing from isolated cardiomyocytes treated with RA identify Tgm2 and Ace1 as RA-responsive genes in cardiomyocytes.Major commentsThe current study is important because it challenges the current dogma on the indirect role of RA in the heart according to which cardiac fibroblasts are the major cell types responsive to RA which acts in an autocrine manner. The new RA-reporter line developed allows lineage tracing of cells responsive to RA with great sensitivity. Overall, this well-designed study reveals important new information on RA protective signaling mechanism in the heart from an impressive amount of in vivo work. While RA signaling is clearly documented in cardiomyocytes during specific windows of heart development, some concerns are raised in the adult heart. In particular, the contribution of cardiomyocytes conferring a protective response after MI should be better documented considering the relatively small fraction responsive to RA. Additional concerns are with the MI model and the use of proper control mice. Major issues that need to be addressed are listed below.

We were pleased to read the overall very positive evaluation by reviewer 1, who found our study to be “important”, “well-designed”, and to contain an “impressive amount of in vivo work”.

1. One concern is with the claim that RA signaling in cardiomyocytes and especially in adult cardiomyocytes, is independent of the epicardium and arises from a local source of RA based on GFP positive cells located away from the epicardium. A relatively small population of cardiomyocytes respond to RA signaling and are detected because of the sensitivity of the new RARE-Cre-ERT2 line generated. The question is what is the contribution of the cardiomyocyte population in the cardioprotective effect of RA after MI? To address this, the responsiveness to RA signaling should be compared to mice with cardiomyocyte-specific or endothelial-specific expression of RA reporters. What is the extent of MI and cardiac remodeling in these 2 lines? Is-there protection in both lines?

We thank reviewer 1 for these comments. ALDH1A2 protein is upregulated in various cell types in the injury and border zones (this study), and in the epicardium (Kikuchi et al., 2011). We cannot formally rule out that MI-induced epicardial-*Aldh1a2* expression contributes to the GFP response but given the proximity of the GFP cells to the ALDH1A2^+^ cells within the injury/border zones, it is likely that the primary response is local. Regarding the number of cardiomyocytes responding to RA after infarction, we would like to point out that GFP activation in our transgenic strain requires a certain level of RA signaling, and it is possible that a larger number of cardiomyocytes respond to RA than can be detected using our reporter strain. This has been added to the discussion of our revised manuscript in lines 434-438.

We also agree that multiple cell-types respond to RA signaling (e.g. PECAM1^+^ cells, αSMA^+^ cells, TropT^+^ cells, see Figure 4E). Deciphering the exact contribution of each cell type to the phenotype observed with ubiquitous ALDH1A would require deleting all three retinoic acid receptor genes (*Rar*a/b/g*)* in a cell type specific manner. Building these strains would take more than a year and we feel that this is beyond the scope of this paper.

After reading this reviewer’s comments, we realize that the title or our paper may have been misleading, as we have not formally proven that RA signaling in cardiomyocytes is responsible for the better recovery after LAD-ligation. We have therefore changed the title to: “Retinoic acid signaling is directly activated in cardiomyocytes and protects mouse hearts from apoptosis after myocardial infarction”.

2. RNA sequencing was performed from cardiomyocytes isolated from embryonic hearts treated with RA to identify RA responsive genes. What was the reason for not performing RNA-seq in adult cardiomyocytes isolated from RARECreERT2 mouse hearts after MI? At minimum, qPCR should be done to assess whether the 2 genes identified (Tgm2 and Ace1) are also up-regulated in adult RARECreERT2 mouse hearts after MI.

Isolating pure cardiomyocyte populations from adult hearts is difficult and typically yields low amounts of viable cells that are often contaminated with cardiac fibroblasts. Identifying a cardiomyocyte-specific response to RA signaling in adults is therefore difficult to achieve. We thus resorted to E18.5 hearts, for which good cardiomyocyte isolation procedures exist.

Regarding the second point, we have now added qPCR data showing that both *Tgm2* and *Ace1* are upregulated in infarct *RARE-Cre-ER^T2^* hearts when compared to sham controls (Figure 6—figure supplement 2C).

3. The legends of whole-mount X-gal or GFP staining are lacking information on the number of embryos used. Please indicate the number of embryos used in all the appropriate Figures to document scientific rigor.

We thank reviewer 1 for pointing this out. The number of embryos analyzed for the whole-mount stainings have now been added to the figure legends.

4. Negative control immunofluorescence with ALDH1A2 antibody should be provided to show specificity of the signal.

ALDH1A2 protein is not detectable in the uninjured heart; and we have added an inset from the sham *RARE-Cre-ER^T2^* heart stained with ALDH1A2 (Figure 4B), which does not display specific labelling. To further confirm the specificity of the ALDH1A2 antibody, we have performed embryonic deletions of *Aldh1a1/a2/a3* be injecting tamoxifen at E13.5 and analyzing at E18.5. Deletion of *Aldh1a* enzymes during late cardiac development leads to a drastic reduction of ALDH1A2 immunoreactivity in the heart when analyzed by IF. Analysis of *Aldh1a2* mRNA levels by qPCR confirms the knockdown efficiency (>90%) using this strategy. These data have been added in Figure 5—figure supplement 1A-C.

5. Figure 4 documents RA signaling in adult RARECreERT2 hearts after sham or MI. Control CreERT2 hearts after sham or MI should be shown in parallel. Also, what control mice were used in Figure 5?

*Aldh1a1^f/f^;a2^f/f^;a3^f/f^* (triple floxed mice, without the Cre transgene) were used as controls and this information has been now added to figure 5 and to the text (lines 286-287). A previous study has shown that activation of the *CAGGCreER* line (JAX No. 004682) (the same line used here) by tamoxifen injection for 10 days followed by MI led to no transient side effects as revealed by echocardiographic analysis (Konstandin et al., 2013) (see lines 284-286). Hence, we believe there is no need for performing MI surgeries on control *CAGGCreER* as these experiments have already been carried out.

6. It is intriguing that 5 doses of tamoxifen only reduce ALDH1A2 by 50%. Please provide an explanation.

The efficiency of Cre-mediated deletion is highly influenced by chromatin structure and silenced genes are known to be more difficult to excise (Long and Rossi, 2009). We performed tamoxifen injections one week prior to MI, a time point when ALDH1A2 protein is not detected (see Figure 4B and Zhou et al., 2011; Kikuchi et al., 2011) and *Aldh1a2* is therefore likely epigenetically silenced. The incomplete excision in the adult heart can also be seen on the genomic level, which shows approximately only 60% of the recombined allele (Figure 5—figure supplement 1D). We would also like to draw the reviewer’s attention to our response to comment 4, where we demonstrate a much higher reduction (>90%) in *Aldh1a2* mRNA levels upon tamoxifen-induced *CAGGCreER* deletion in late cardiac development, when *Aldh1a2* is actively expressed.

7. The transverse sections of control and RAK ko hearts after MI are atypical as they do not show the expected loss of myocardial tissue especially after 6 days of MI as shown in Figure 4B. Longitudinal sections or cross-sections at different levels should be shown to document the extent of the MI in Figure 5.

Analyzing the loss of myocardial tissue after MI on transverse sections is a standard method that has been implemented in numerous publications (e.g. Gao et al., 2019). In our study, we analyzed 6 transverse sections per mouse, with each section being spaced by 40 µm. A significant and highly representative portion of the infarct zone in operated mice has therefore been analyzed. As to why the expected loss of tissue is less than expected by reviewer 1 is not clear, but this may be due to surgical procedures. We would like to point out that all surgeries were performed by the same researcher using the same conditions for both control and *RAKO* mice. We are therefore confident that the observed increase in infarct size is due to the loss of RA signaling rather than experimental variations.

Analyzing infarct areas on longitudinal sections is not possible in the hearts collected since we removed the apices for RNA extraction.

8. Because it is challenging to obtain reproducible infarcts after MI surgery, the number of mice should be increased to at least 10 in each group.

We have increased the number of mice analyzed to 10 per group in our revised version of the manuscript, as suggested by reviewer 1.

9. Are the survival rates of control and RAK mice after MI identical? With such a difference in fibrosis, one would expect that control mice have a longer life span than RAK ko after MI.

In Figure 5—figure supplement 2D we now document the survival rates of RAKO at 6 days post MI. As predicted by reviewer 1, *RAKO* mice exhibit increased early death when compared to controls (4 out of 14 *RAKO* mice died before 6 days, while no early death is documented in control mice).

10. What is the effect of BMS493 in sham or MI operated mice? Inactivation of RA signaling with BMS493 should reduce the infarct size of RAK ko mice after MI.

Our data suggest that RA signaling has a protective effect after myocardial infarction and BMS493 treatment would not ‘rescue’ the increased infarct in RAKO mice. In addition, BMS493 has a relatively short half-life and mice would have to be injected several times per day to efficiently block RA signaling.

Minor points1. Line 92: add reference 22 from Guleria 2011.

This has been corrected.

2. Please run a spell check for typos.

This has been performed.

Reviewer #1 (Significance (Required)):The study is significant technically because a new mouse line allowing RA lineage tracing at specific times of development and in multiple tissues was developed. The conceptual novelty is with the finding that a population of cardiomyocytes is responsive to RA signaling at late stages of embryonic development and also in the injured adult heart. As an expert in heart failure mechanisms, I believe the current study is potentially highly significant for the cardiac field.

We were pleased to read that Reviewer 1 found our study to be “technically significant” and to be “potentially highly significant for the cardiac field”.

Referees cross-commentingI totally agree with the comments of Reviewer 2 with respect to issues with the MI experiments and the lack of quantitative analysis using immunofluorescence to assess RA signaling in general. I am also in agreement with the issues raised by Reviewer 3.Reviewer #2 (Evidence, reproducibility and clarity (Required)):A novel cell line was developed to monitor retinoic acid signaling (RA) that is useful in multiple cell lines. The activation of retinoic acid signaling was studied during embryogenesis and post-MI. There is shown that RA signaling is activated mainly in fibroblasts post-MI and that downregulation of RA leads to allegedly larger infarcts with greater scar is shown. Classic responses to select RA downstream effectors is shown. The studies are well written. Technical issues include the lack of a risk area assessment in the infarct studies, making the assertion that infarct size is increased an unsubstantiated assertion. Cellular studies are performed with exogenous RA that show that activation of RA gene programs occurs.

This reviewer is correct that our study did not include a risk assessment analysis. We disagree however with his/her opinion that our claim of increased infarct size in *Aldh1a* mutant animals is unsubstantiated. Indeed, our measurements performed on sections show a statistically significant increase in infarct size in animals in which *Aldh1a1/2/3* have been deleted.

Reviewer #2 (Significance (Required)):Though stated in the introduction; the clear advantage of the new cell line should be better documented by both direct comparison studies and an improved discussion. The indicator appears to be all or none; the extent of activation of RA signaling, the amount of RA needed to turn a cell "positive" and the durability of the response remain unclear. The data during embryogenesis are all qualitative fluorescence studies, a relative limitation.

We would like to thank reviewer 2 for these helpful comments. Unfortunately, a quantitative assessment of the amount of atRA required to activate RA signaling in vivo is difficult to perform*.* As a solution, we have addressed these issues using primary cardiomyocytes isolated from *RARECreER^T2^*; *mTmG* E18.5 hearts. Briefly, by treating cells with 10 nM of atRA and then performing live cell imaging, we could detect first GFP^+^ cells after 28 hours of RA treatment, and that the levels of GFP^+^ cells peaked around 48 hours. We then treated cells with different doses of atRA (i.e. 0.1 nM-1000 nM) for 48 hours (length of treatment based on time course experiment) and counted the number of GFP^+^ cells. Cardiomyocytes responded to RA signaling in a dose dependent fashion (Figure 2I). Moreover, as shown in Figure 2—figure supplement 1D, we could detect GFP^+^ cells with RA concentrations as low as 0.1 – 1 nM, which are physiologically relevant. These quantitative data provide valuable information on the doses of RA required to ‘turn’ cardiomyocytes ‘positive’.

We are not sure we understood the question regarding the “durability” of the response correctly. *Cre*induced recombination of the mTmG transgene using the *RARE-CreER^T^*^2^ line leads to a permanent activation of GFP.

The potential occurrence of select receptor activation by RA should be discussed, the cell line clearly only detects RARb. Also, non-receptor interactions of RA should be discussed.

The *RARECreER^T2^* line utilizes the retinoic acid response element (RARE) from the *Rarb* gene, a faithful direct target gene of RA signaling, that is the most commonly employed strategy for generating RA reporter mouse lines (Rossant et al., 1991; Dolle et al., 2010; Bilbija et al., 2012). However, we do realize that activation of the *Rarb* RARE may not reflect RA signaling in its complete capacity, nor does it account for non-receptor interactions, as correctly pointed out by reviewer 2. This is a limitation of our line which has now been addressed in lines 384-387 of the discussion.

Referees cross-commentingAgree with the comments of Reviewers 1 and 3; especially major comment 1 of Reviewer 1.Reviewer #3 (Evidence, reproducibility and clarity (Required)):Summary:The paper tests a new inducible murine transgenic retinoic acid-reporter line that allows lineage tracing experiments for cells in which retinoic acid occurs. This is used to show that retinoic acid directly acts on cardiomyocytes during mid-late gestation and interestingly also occurs after myocardial infarction in the adult. An appropriate set of controls were used and impressive that there is a strong inhibition of reporter when the RAR inverse agonist BMS493 is applied. Overall the studies are logical and provide a direct test of well formulated ideas.

We were happy to read the overall positive evaluation by reviewer 3 who found our study to be logical and to provide a direct test of well formulated ideas.

Major comments:1. Aldh1a1 and Aldh1a3 transcripts increase with infarction. What is their protein distribution compared to Aldh1a2? This is particularly important because the deletion of enzyme studies results in only 50% reduction in ALDH1A2 protein (Figure 5C) but a near 70% decrease of Aldh1a1 and Aldh1a2 mRNA levels. In addition, quantitation of change in protein expression by fluorescence intensity will be highly inexact. If this is the best approach some attempt to demonstrate how this relates to real levels needs to be made.

We agree that measuring ALDH1A2 protein-levels by IF is not very reliable, and we have removed Figure 5C form the paper.

We had previously tested several commercial antibodies for ALDH1A1 and ALDH1A3, but found them to be unsatisfactory. We have now tested new antibodies but did not observe ALDH1A1 or ALDH1A3 immunoreactivity in the MI heart by IF. Hence, due to technical difficulties, we were not able to provide data on the distribution of ALDH1A1 and ALDH1A3 protein in the heart after MI.

2. It was proposed that regulation of Tgm2 by RA was direct from its action within 9 hours however given measurable regulation of transcript can occur in 2-3 hours this argument is not strong.

This is an excellent point brought up by reviewer 3. Indeed, the claim that the activation of *Tgm2* by RA may be direct based on the 9 hours RA treatment data could be strengthened. To further test whether *Tgm2* and *Ace1* are direct or indirect RA targets, we performed experiments where RA treatment was reduced to 3 hours, and in which the protein synthesis inhibitor cycloheximide was supplemented to the media to further limit indirect activation of *Tgm2* and *Ace1.* As shown in figure 6 Figure supplement 2D, no modulation of *Tgm2* or *Ace1* expression is observed upon 3 hour RA treatment, both with and without cycloheximide. *Rarb* expression is still upregulated with only 3 hour RA treatment. Altogether, this data suggests that *Tgm2* and *Ace1* are indirect RA targets. This new data is addressed in lines 352-359 of the Results section and in lines 447-448 of the discussion.

Minor comments:1. On line 36 it is described that oxidation of retinaldehyde to retinoic acid is considered the rate-limiting step. What is the reference for this? In the past it was considered the oxidation of retinol to retinal e.g. Wolf G. The regulation of retinoic acid formation. Nutr Rev. 1996;54:182-4.

We agree with reviewer 3 that this is a controversial statement and we have removed it from the manuscript.

2. What might account for differences with the Rossant mouse retinoic acid reporter mouse in the limb buds i.e. Rossant showing low/absent reporter in the limbs at E8.5?

The *RARE-LacZ* line used in Rossant et al. 1991 documents acute activation of RA activity and does not permanently label cells. The *RARECreER^T2^* line, coupled with the *Rosa26 LacZ* or *mTmG* reporters, permanently labels RA-responsive cells and is thus suitable for cell lineage analysis. We therefore believe that the labelling of the limbs with our *RARECreER^T2^* line is a result of RA activation in limb progenitors at an early stage (e.g. E6.5 or E7.5), which by E8.5 have migrated into the limb buds and, due to the permanent nature of the reporter activation, continue to express *β-gal*/GFP. Testing this hypothesis would require specific experiments that are beyond the scope of this paper.

3. On line 414-415 the discussion suddenly jumps to discussion of the RAS pathway without any explanation of this.

We agree that the transition was very abrupt, and we thank reviewer 3 for pointing this out. This has been corrected.

Reviewer #3 (Significance (Required)):Nature and significance:The results test a new inducible murine transgenic retinoic acid-reporter line valuable for future developmental studies of retinoic acid signaling. The study also provides further evidence for the role of retinoic acid in repair of heart after myocardial infarction.The work is an important step forward from present knowledgeAudience: Developmental biologists and cardiac physiologistsMy expertise: retinoic acid signalingReferees cross-commentingAgree with all comments

References:

Konstandin, M.H., Toko, H., Gastelum, G.M., Quijada, P., De La Torre, A., Quintana, M., Collins, B., Din, S., Avitabile, D., Völkers, M., et al. (2013). Fibronectin is essential for reparative cardiac progenitor cell response after myocardial infarction. Circ. Res. 113, 115–125.

Kikuchi, K., Holdway, J.E., Major, R.J., Blum, N., Dahn, R.D., Begemann, G., and Poss, K.D. (2011). Retinoic Acid Production by Endocardium and Epicardium Is an Injury Response Essential for Zebrafish Heart Regeneration. Dev. Cell 20, 397–404.

Gao, F., Kataoka, M., Liu, N., Liang, T., Huang, ZP., Gu, F., Ding, J., Liu, J., Zhang, F., Ma, Q., Wang, Y., Zhang, M., Hu, X., Kyselovic, J., Hu, X., Pu,WT., Wang, J., Chen, J. and Wang D. (2019). Therapeutic role of miR-19a/19b in cardiac regeneration and protection from myocardial infarction. Nature Communications 10, 1802.

Long, M.A., and Rossi, F.M.V. (2009). Silencing inhibits Cre-mediated recombination of the Z/AP and Z/EG reporters in adult cells. PLoS One 4(5), e5435.

Zhou, B., Honor, L.B., He, H., Qing, M., Oh, J.H., Butterfield, C., Lin, R.Z., Melero-Martin, J.M., Dolmatova, E., Duffy, H.S., et al. (2011). Adult mouse epicardium modulates myocardial injury by secreting paracrine factors. J. Clin. Invest. 121, 1894–1904.